# Inflammation-Related Carcinogenesis: Lessons from Animal Models to Clinical Aspects

**DOI:** 10.3390/cancers13040921

**Published:** 2021-02-22

**Authors:** Futoshi Okada, Runa Izutsu, Keisuke Goto, Mitsuhiko Osaki

**Affiliations:** 1Division of Experimental Pathology, Faculty of Medicine, Tottori University, Yonago 683-8503, Japan; b17m2007y@edu.tottori-u.ac.jp (R.I.); k.goto.san@tottori-u.ac.jp (K.G.); osamitsu@tottori-u.ac.jp (M.O.); 2Chromosome Engineering Research Center, Tottori University, Yonago 683-8503, Japan; 3Division of Gastrointestinal and Pediatric Surgery, Faculty of Medicine, Tottori University, Yonago 683-8503, Japan

**Keywords:** inflammation-related carcinogenesis, animal models, therapeutic apheresis

## Abstract

**Simple Summary:**

In multicellular organisms, inflammation is the body’s most primitive and essential protective response against any external agent. Inflammation, however, not only causes various modern diseases such as cardiovascular disorders, neurological disorders, autoimmune diseases, metabolic syndrome, infectious diseases, and cancer but also shortens the healthy life expectancy. This review focuses on the onset of carcinogenesis due to chronic inflammation caused by pathogen infections and inhalation/ingestion of foreign substances. This study summarizes animal models associated with inflammation-related carcinogenesis by organ. By determining factors common to inflammatory carcinogenesis models, we examined strategies for the prevention and treatment of inflammatory carcinogenesis in humans.

**Abstract:**

Inflammation-related carcinogenesis has long been known as one of the carcinogenesis patterns in humans. Common carcinogenic factors are inflammation caused by infection with pathogens or the uptake of foreign substances from the environment into the body. Inflammation-related carcinogenesis as a cause for cancer-related death worldwide accounts for approximately 20%, and the incidence varies widely by continent, country, and even region of the country and can be affected by economic status or development. Many novel approaches are currently available concerning the development of animal models to elucidate inflammation-related carcinogenesis. By learning from the oldest to the latest animal models for each organ, we sought to uncover the essential common causes of inflammation-related carcinogenesis. This review confirmed that a common etiology of organ-specific animal models that mimic human inflammation-related carcinogenesis is prolonged exudation of inflammatory cells. Genotoxicity or epigenetic modifications by inflammatory cells resulted in gene mutations or altered gene expression, respectively. Inflammatory cytokines/growth factors released from inflammatory cells promote cell proliferation and repair tissue injury, and inflammation serves as a “carcinogenic niche”, because these fundamental biological events are common to all types of carcinogenesis, not just inflammation-related carcinogenesis. Since clinical strategies are needed to prevent carcinogenesis, we propose the therapeutic apheresis of inflammatory cells as a means of eliminating fundamental cause of inflammation-related carcinogenesis.

## 1. Introduction

Inflammation, especially in the chronic form, is the definite cause of tumor development and progression, and it is referred to as “inflammation-related carcinogenesis”. The causes of chronic inflammation include pathogens such as viruses, bacteria, and parasites or irritants that are inhaled (asbestos, tobacco smoke, ambient particulate matter, and nano-sized materials) or some types of autoimmune reaction. It is important to understand that the pathogens causing inflammatory diseases are unrelated to each other, whereas the essential pathological feature in common, i.e., sustained inflammation.

Inflammation begins initially as infiltration of activated phagocytes/lymphocytes, which then stimulates stromal reaction mainly composed of fibroblasts. In principle, continuous generation of genotoxic reactive oxygen species (ROS), reactive nitrogen species (RNS), and reactive sulfur species (RSS) by these activated inflammatory cells may injure or affect normal healthy cells [1]. Surviving cells undergo compensatory proliferation under the influence of soluble factors released from infiltrated inflammatory cells, which lead to DNA damage/accumulation of genetic alterations and stimulation of the uptake of internal/external carcinogens. Inflammation is thus the most effective intrinsic factor required to repeatedly maintain the states of “tissue damage”, “genetic instability”, and “growth stimulation”. Moreover, epigenetic alterations such as aberrant DNA hypermethylation are involved in the inflammation-related cancers [2]. From these facts, we understand that inflammation itself promotes genetic alterations (initiator), hyperproliferative response (promoter), and DNA methylation at the same time in the inflammatory sites. All these events from the critical and essential steps for carcinogenesis. Based on this information, the inflammatory environment can generally be understood as the “carcinogenic niche” [3].

Inflammation-related carcinogenesis is a long-term process, involving the conversion from normal cells to tumorigenic cells through precancerous lesions and, thereafter, to acquired malignant phenotypes. Various models of inflammation-related carcinogenesis that mimic human etiology have been developed; both in vitro and in vivo models are available to utilize. The in vitro model is suitable for rapid analysis of gene function and soluble mediators in cultured cells. However, the in vitro model targets specific stages of tumorigenesis. In contrast, in vivo models require long observation periods but are suitable for using animals to reproduce inflammation-related carcinogenesis actually formed in humans, identify carcinogenic mechanisms, and validate preventative strategies.

Inflammation has been shown to affect tumor cells as well as tumorigenic conversion. In addition to the direct involvement of inflammatory cells infiltrating tumor tissue, inflammation associated with metabolism-related diseases such as obesity and diabetes has recently been shown to accelerate tumor malignancies. However, these topics are beyond the scope of this paper and have been reported in some excellent reviews [3,4,5,6,7]. This review focuses on organ-specific animal models that promote inflammation-related carcinogenesis and suggests the clinical strategies needed to prevent carcinogenesis by identifying common causes from animal models.

## 2. Materials and Methods—Data Sources and Search Strategy

Using PubMed’s database, a systematic review of the published research papers on experimental animal models of inflammation-related carcinogenesis was performed. The search on PubMed (https://pubmed.ncbi.nlm.nih.gov/) was limited to those studies that were published from inception to 18 July 2020 with the following search terms: “inflammation” and “related” and “carcinogenesis” and “animal” and “model”. A PubMed database search identified 389 articles. Additionally, 11 articles written in languages other than English were excluded. Three thousand and ninety-six articles were added by the reference list from retrieved articles or hand searches. Out of a total of 3474 articles, 472 articles that clearly described the inflammatory response were assessed.

## 3. Inflammation-Related Carcinogenesis Model of Each Organ

### 3.1. Oral Cavity and Tongue Cancer

The incidence of oral cancer is high particularly in developing countries, where up to 25% of all cancers occur in young men. Smoking, alcohol, betel quid chewing, exposure to human papillomavirus, and inflammation have been reported as common causes of oral cancer [8].

#### 3.1.1. Alcohol

There were two typical experiments to assess long-term alcohol consumption. One experiment involved the administration of 10% alcohol to rats as drinking water. Spontaneous oral cancer occurred in approximately 3% rats that received only water, whereas it occurred in approximately 10% rats that received 10% alcohol. In another experiment, mother rats received alcohol before mating, and their offspring continued to receive alcohol for the rest of their lives. When the control rats received water, approximately 5% of the rats developed cancer. Around 30–40% offspring developed cancer during their lifetimes [9].

#### 3.1.2. Chemical Carcinogenesis Model

7,12-Dimethylbenz[a]anthracene (DMBA) and alcohol intake stimulated hamster cheek pouch carcinogenesis mediated by lipid peroxidation in the inflammatory environment [10].

4-Nitroquinoline-1-oxide (4-NQO) induced tumors in the tongue. When mice were treated with 4-NQO and with ethanol in drinking water, they developed mast cell infiltration and had increased expressions of the inflammatory mediators, 5-lipoxygenase and cyclooxygenase-2 (COX-2), and, thereafter, developed oral dysplasia and squamous cell carcinoma [11].

#### 3.1.3. Genetically Modified Host Model

Overactivation of the inhibitor of nuclear factor κ-B kinase (IKK) complex caused the activation of nuclear factor κ-B (NF-kB), leading to severe inflammation. In the oral epithelia of IKK subunit β (IKKβ) transgenic mice, produced in persistent lichenoid inflammation due to neutrophil, macrophage, and B cell infiltration, spontaneous oral squamous cell carcinomas were formed. Oral carcinogenesis in IKKβ transgenic mice could not be observed in IKKβ/nude mice (lacking T cells) or IKKβ/SCID mice (lacking T, B, and natural killer (NK) cells), which indicated that oral cancer was caused by inflammatory cells other than T, B, and NK cells. IKKβ transgenic mice also showed inflammation of the forestomach epithelia and developed squamous cell carcinomas [8].

Homozygous deletion of Toll-like receptor 2 (TLR2^−/−^) using 4-NQO in mice increased the number of tongue-infiltrating M2 macrophages and resulted in the development of tongue cancer [12]. Gene polymorphisms of TLR2 were associated with risk and survival in human oral cancer [12].

### 3.2. Esophageal Cancer

Esophageal adenocarcinoma developed from a premalignant disease, intestinal-like columnar metaplasia, also known as Barrett’s esophagus, characterized by the replacement of stratified squamous epithelium in the distal esophagus linked to reflux esophagitis [13]. The link between inflammation and esophageal cancer is well-established; in particular, interleukin (IL)-1β is overexpressed in Barrett’s esophagus, and polymorphisms in the IL-1β gene are associated with Barrett’s esophagus [13].

#### 3.2.1. Esophagoduodenal Anastomosis Model

Esophageal duodenal anastomosis formed in rats is the model for causing duodenoesophageal reflux and spontaneously developing esophageal adenocarcinoma. In this model, epithelial erosions and ulcers with inflammatory cell infiltration (lymphocytes, eosinophils, and macrophages) and inflammatory cytokine induction (IL-1β, IL-6, and IL-8) are observed at the esophageal–duodenal junction [14].

#### 3.2.2. Diet

The development of esophageal adenocarcinoma was accelerated by inflammation induced by treatment with a high-fat diet and bile acids [15]. The predominant inflammatory exudative cell type of esophagitis that leads to dysplasia and cancer in the CXCR4^+^ cell populations, which includes neutrophils and CD3^+^ T cells but excludes B, NK, and endothelial cells and fibroblasts. The accumulation of CXCR4^+^ cells is also observed in patients with esophageal adenocarcinoma [15].

#### 3.2.3. Genetically Modified Host Model

A transgenic mouse model in which the human IL-1β gene is overexpressed in the esophagus reflects the pathology underlying the progression toward Barrett’s esophagitis in humans, i.e., from low-grade and high-grade dysplasia to esophageal adenocarcinoma [15].

### 3.3. Lung Cancer

The frequency of chemical carcinogenesis to the lungs is affected by strain differences in mice. The C57BL/6 strain is considered to be carcinogenic-tolerant, while the BALB/c strain is moderately sensitive. The most sensitive strain is the A/J mice, which is commonly used in lung carcinogenesis models. The reason for the highest sensitivity of A/J mice to the development of lung tumors is that they have the K-ras intron 2 polymorphism and, thus, undergo activation of the K-ras oncogene. The activated K-ras oncogene is a driver of lung carcinogenesis and is widely used in inflammation-related carcinogenesis models. However, the spontaneously developing lung tumors in A/J mice are always adenomas [16].

#### 3.3.1. Inhaled Particle Model

Libby amphibole (LA) asbestos, which mainly contains asbestos and tremolite, comes in a variety of crystalline forms, from asbestos to needle-like or prismatic. Occupational exposure to this LA is known to cause localized and diffuse pleural thickening, asbestosis, lung cancer, and mesothelioma. In a rat model, the intranasal inhalation of LA caused the spontaneous development of alveolar epithelial hyperplasia and bronchiolar/alveolar adenoma or lung carcinoma by fiber-induced chronic inflammation with massive infiltration of neutrophils and macrophages [17]. It has also been confirmed that the inhalation of amosite asbestos causes lung carcinogenesis due to chronic inflammation [17].

Silicosis develops from occupational exposure to crystalline silica, which is associated with lung cancer. In a rat model, intratracheal instillation of crystalline silica dust caused the spontaneous development of lung adenocarcinoma and squamous cell carcinoma by inducing irreversible silicotic granulomas, which consist of aggregates of activated macrophages that phagocytose silica, induce collagen deposition and recruit lymphoid cells. Chronic silicosis is caused by inflammatory mediators such as interleukins, tumor necrosis factor (TNF)-α, transforming growth factor (TGF)-β, and reactive oxygen species (ROS)/reactive nitrogen species (RNS) [18].

The intrapulmonary administration of titanium dioxide (TiO_2_) through spray to human c-Ha-ras proto-oncogene transgenic rats treated with the carcinogen di(2-hydroxypropyl)nitrosamine caused an accumulation of TiO_2_ aggregates in macrophage inflammatory protein 1 α (MIP1α)-stimulated alveolar macrophages, and the rats developed hyperplasia and adenomas [19].

#### 3.3.2. Cigarette Smoke

Continuous exposure of mice to mainstream cigarette smoke in an exposure chamber resulted in the development of lung adenomas and cancers with chronic inflammation, emphysema, and alveolar epithelial hyperplasia [20].

#### 3.3.3. Virus Lysate

The lysate of non-typeable *Haemophilus influenzae* colonizes the airways of patients with chronic obstructive pulmonary disease (COPD) or heavy smokers. The aerosolized lysate itself induced inflammatory cell exudate (neutrophils, macrophages, and CD8^+^ T cells) found in COPD patients, followed by the formation of lung cancer in activated K-ras transgenic mice [21].

#### 3.3.4. Chemical Carcinogenesis Model

A common lung carcinogenesis model is formed by the administration of nicotine-derived nitrosamine ketone (NNK), 4-(methylnitrosamino)-1-(3-pyridyl)-1-butanone, to A/J mice. The chronic intranasal administration of lipopolysaccharide (LPS), a major proinflammatory glycolipid component of the outer membrane of Gram-negative bacteria, promotes NNK-induced lung tumorigenesis [16]. LPS is ubiquitous in the environment, especially in cigarette smoke, and a person who smokes a pack of cigarettes a day is exposed to approximately 2.5 μg of LPS. LPS-induced inflammatory and pathological changes are similar to the features of human COPD, including the recruitment of activated macrophages [16].

Most of the experimental adenocarcinomas exhibited mutations in K-ras, either in codon 12 or in codon 61. Lung adenocarcinoma developed in mice treated with N-nitrosodimethylamine (NDMA) alone showed A-to-G transition (Q61R) at K-ras codon 61, whereas mice treated with NDMA and oropharyngeal administration of silica showed G-to-A transition (G12D) in codon 12 [22]. Therefore, among the K-ras mutation profiles of lung cancer, the Q61R to G12D mutations can occur in an inflammatory environment. 

Benzo[a]pyrene (B[a]P) is present in tobacco metabolites and is metabolized into epoxide, which induces DNA adducts and causes mutations due to ROS generation, which can together accelerate the process of lung tumorigenesis. C57BL/6 mice exposed to B[a]p with intratracheal LPS administration formed more lung tumors than mice exposed to B[a]p alone [23].

A mouse model of non-small cell lung carcinoma was generated using a halogenated *N*-nitroso-trischloroethylurea and by the intranasal instillation of LPS in A/J mice [23]. In the model, LPS-induced inflammation has been shown to increase the malignancy of the developing tumors [24].

Chronic inflammation induced by continuous administration of the inflammatory agent butylated hydroxytoluene promotes lung carcinogenesis initiated by the carcinogen of 3-methylcholanthrene [25].

#### 3.3.5. Genetically Modified Host Model

The homozygous conditional deletion of mitogen-induced gene 6 (Mig-6^d/d^) and mutated K-ras (K-Ras^G12D^) occurs in transgenic mouse (Mig-6^d/d^K-Ras^G12D^) in which a signaling molecule, Mig-6, is deleted and K-ras (G12D) is activated in lung Clara cells. These mice spontaneously develop lung adenoma and atypical adenomatous hyperplasia by inducing inflammatory response [26].

Transgenic mice with the oncogenic activated form of signal transducers and activators of transcription 3 (Stat3C) stimulate alveolar type II epithelial cell growth, inflammatory cytokine/chemokine expression, and immunosuppressive infiltration of macrophages and lymphocytes, resulting in the development of a lung adenocarcinoma [27].

### 3.4. Stomach Cancer

The incidence of spontaneous gastric cancer in rats is extremely low, with only one gastric adenocarcinoma found at autopsy in 33,000 rats (0.003%) [28]. The main factor in the gastric carcinogenesis model is infection of the oncogenic bacterium *Helicobacter pylori* (*H. pylori*) infection [29].

#### 3.4.1. Helicobacter pylori Model

The conclusive evidence that *H. pylori*-induced chronic inflammation causes gastric cancer is validated by the fact that eradication suppresses the *H. pylori*-related inflammatory response and, as a result, inhibits gastric carcinogenesis [29]. Changes in gastric epithelial cells infected with *H. pylori* have been shown to differ between clinical and animal models. In humans, *H. pylori* infection results in two precancerous lesions: intestinal metaplasia that develops in chronic inflammatory conditions and antispasmodic polypeptide expression metaplasia (SPEM) that develops after parietal cell loss. However, mouse infections do not cause intestinal metaplasia, and most exhibit SPEM [30].

#### 3.4.2. Spontaneous Carcinogenesis Model

Exceptionally, primates and Mongolian gerbils are the only animals that undergo induction of gastritis and adenocarcinoma with *H. pylori* infection without carcinogens [31]. *H. pylori*-related gastric cancers occur in Mongolian gerbils but seldom in mice. Comparing the inflammatory response in the gastric mucosal epithelium infected with *H. pylori*, mucosal destruction and proliferation occurred in the Mongolian gerbils but scarcely in mice. The phenomenon of the occurrence of gastric carcinogenesis due to *H. pylori* infection is observed only in Mongolian gerbils, which is thought to be due to the proliferative response of gastric epithelial cells to the trophic factor gastrin [31].

#### 3.4.3. Carcinogen-Assisted Carcinogenesis Model

In a Mongolian gerbil model, *H. pylori* is delivered intragastrically after initiating using a carcinogen, *N*-methyl-*N*-nitrosourea (MNU). *H. pylori* infection resulted in active chronic gastritis, characterized by edema and erosions, atrophy, hyperplasia, and marked infiltration of inflammatory cells. The histological types of gastric cancer developed by *H. pylori* and MNU caused poorly differentiated, signet-ring cells and well-differentiated adenocarcinoma [29].

The general perception is that *H. pylori* is thought to infect only primates and Mongolian gerbils. However, it has become possible to induce gastric carcinogenesis in mice by selecting a suitable strain for infecting *H. pylori* in mice. The *H. pylori* Sydney strain-1 (SS1) adapted for the mouse infection was isolated, which revealed that there are mouse strain differences in the intensity of the inflammatory response induced by the infection [11]. BALB/c mice develop mild gastritis, while C57BL/6 mice exhibit extensive gastritis. This may be reflected in C57BL/6 mice inducing a Th1 response and BALB/c mice inducing a Th2 response to bacterial infection. C57BL/6 mice infected with *H. pylori* SS1 after MNU treatment cause chronic atrophic gastritis and metaplasia with infiltration of inflammatory cells, eventually developing gastric adenomas and adenocarcinomas [32].

#### 3.4.4. Helicobacter felis Model

Compared to *H. pylori*, mice infected with the murine pathogen *Helicobacter felis* (*H. felis*) are more likely to develop severe inflammation and undergo damage to the gastric mucosa, similar to human pathobiology. Infecting C57BL/6 mice with the SS1 strain, a mouse-adapted *H. pylori* strain, without a carcinogen causes chronic SS1 colony formation and hypertrophy in the stomach, but they do not develop dysplasia and carcinoma. C57BL/6 mice infected with *H. felis* show histological changes, as well as human infections—namely, severe chronic gastritis, atrophy, metaplasia, and dysplasia—without the need for a carcinogen. In addition, invasive adenocarcinomas are spontaneously formed in these mice [33].

To develop a *Helicobacter* infection-dependent formation of atrophic gastritis and gastric carcinogenesis in mice, T cells are required. Both Rag2^−/−^ mouse (deficient in T, B, and NK cells but have myeloid-derived suppressor cells) and TCRβδ^−/−^ mice (T cell-deficient) show no detectable epithelial changes or parietal cell loss when infected with *H. pylori* or *H. felis* [34].

#### 3.4.5. Chemical Carcinogenesis Model

WBN/Kob rats develop long-lasting diabetic symptoms with age without a carcinogen. Alloxan, a nongenotoxic diabetogenic chemical, induced the growth of squamous epithelium in the tongue, esophagus, and forestomach and delayed the onset and acceleration of diabetes due to injury of insulin-producing β-cells through ROS generation. Alloxan-induced chronic inflammation mainly consisted of neutrophils and monocytes in rat forestomach, and the animals developed well-differentiated squamous cell carcinoma [35].

#### 3.4.6. Gastrojejunostomy Model

Gastric stump resection was developed as a model for spontaneous gastric carcinogenesis. Increased carcinogenic frequency depends on the reflux fluid passing through the remaining stomach and the length of the follow-up period. When two-thirds of the glandular stomach is surgically resected in rats and anastomosed to the proximal jejunum, the remaining stomach repeatedly elicits inflammation and tissue-damaging responses induced by duodenal gastric reflux disease. Gastric cancer occurred at approximately nine months after surgery [36].

#### 3.4.7. Genetically Modified Host Model

The homozygous knock-in substitution of a glycoprotein 130 (gp130^Y757F/Y757F^) mouse carries a mutated gp130 cytokine receptor signaling subunit that cannot bind the negative regulator of cytokine signaling SOCS3 and hyperactivates STAT1 and STAT3 signaling. The activated signals then transactivate the downstream target IL-11, promoting chronic gastric inflammation and the associated tumorigenesis. gp130^Y757F/Y757F^ causes the development of those gastric cancer-accompanying hyperplasia, with histological features reminiscent of those intestinal-type and metaplastic gastric tumors in humans [37].

The stomach-specific expression of human IL-1β in transgenic mice leads to gastric inflammation and cancer by the early recruitment of myeloid-derived suppressor cells, while T and B cells are not needed for this phenomenon. IL-1β activates NF-κB and increases the production of IL-6 and TNF-α. Bone marrow-derived suppressor cells that cooperate with these inflammatory cytokines cause chronic gastritis and eventually develop high-grade dysplasia or adenocarcinoma in mice over one year [38]. Carriers of IL-1β polymorphisms are at risk for human gastric cancer [39].

### 3.5. Liver Cancer

Chronic infections with hepatitis B (HBV) and C (HCV) viruses, exposure to aflatoxin, alcoholic damage, obesity, and genetic disorders are reported to be involved in the development of hepatocellular carcinoma (HCC); however, the etiology remains unknown in almost 50% of HCC patients. Recently, nonalcoholic fatty liver disease (NAFLD) and its severe form nonalcoholic steatohepatitis (NASH) are associated with an increased risk of HCC [40].

#### 3.5.1. HBV Model

More than 50% of all HCC cases globally are attributed to persistent HBV infection-associated liver cirrhosis. HBV infection varies worldwide, especially high in Southeast Asia and sub-Saharan Africa, and more than 70% of HCC patients are positive for the hepatitis B surface antigen (HBsAg) [41].

Mammalian hepadnaviruses closely related to HBV, including woodchuck hepatitis virus (WHV), ground squirrel hepatitis virus (GSHV), Richardson squirrel hepatitis virus (RSHV), and arctic squirrel hepatitis virus (ASHV), induce acute and chronic hepatitis and spontaneous liver tumorigenesis in their hosts. In particular, the tumorigenic process in WHV-infected woodchucks appears to be quite similar to HBV-related hepatocarcinogenesis in humans, despite the absence of cirrhosis [42].

Woodchuck colonies kept in the zoo were virtually infected with WHV after birth and became chronic WHV carriers, with approximately 15% of them having HBsAg-positive sera and developing HCC through chronic active hepatitis. Infiltrated inflammatory cells include all types of inflammatory cells [42]. This finding shows evidence that a woodchuck hepatitis virus that is a similar to human HBV may act as causative agents of hepatitis in species other than human beings.

The HBsAg-transgenic mouse develops progressive hepatic damage due to chronic hepatitis until the onset of HCC as a result of accumulation of the HVB virus envelope protein (HBsAg) in hepatocytes [43]. HBsAg in hepatocytes induces liver regeneration due to induction of inflammation and the associated ROS, which impairs their ability to replicate and allows the activation of hepatic stem cells [44]. The inflammatory response of the liver in HBsAg transgenic mice includes activated resident immune cells, perivascular fibrosis, and varying degrees of steatosis but not advanced fibrosis or cirrhosis and then, the development of HCC [43].

The HBV large envelope-transgenic mouse develops chronic hepatitis, finally leading to the onset of HCC that mimics human chronic active hepatitis-related hepatocarcinogenesis. The HBV-envelope transgenic mice show a stimulation of Kupffer cells and induction of hepatic oxidative DNA damage [43].

#### 3.5.2. HCV Model

There are four different HCV transgenic mouse lines, which include lines carrying the HCV genome, core genes, envelope genes, or nonstructural genes under the same transcriptional control element. Among these mice, only the transgenic mice carrying the core gene mainly develop HCC [45].

Transgenic mice carrying the HCV genome develop less frequently spontaneous HCC and, like the other HCV transgenic mouse models, do not form cirrhosis. The frequency of carcinogenesis depends on the genetic background of the mouse, with C57BL/6 mice strain being the most sensitive and the FVB mice strain being the most resistant [46].

Among the HCV transgenic mice, mice carrying the HCV core gene developed hepatic steatosis, along with lymphoid follicle formation and bile duct damage, at the early stage and then developed HCC. In mice, lipid droplets are found on benign adenoma cells but are rarely observed on HCC. The phenomenon of inverse correlation between the accumulation of lipid droplets and the degree of differentiation of HCC is similar to that observed in patients with chronic HCV [45].

Transgenic mice carrying the envelope gene (E1 and E2) rarely develop HCC despite the high expression levels of both proteins; however, the administration of carcinogens does not change the tumorigenic frequency but increases the size of the tumor [45,46].

Transgenic mice carrying the HCV nonstructural gene develop steatosis, but the frequency of carcinogenesis is extremely low. Treatment with various chemical irritants increases carcinogenic susceptibility [45,46].

Other than humans, the only animal species fully susceptible to HCV infection is the chimpanzee. HCV infection in chimpanzees demonstrated almost all features of human HCV, including hepatocarcinogenesis, with the exception of no cirrhosis and minimal fibrosis. This finding supports the existence of clinical cases of HCV-related HCC occurring in patients without cirrhosis, albeit at a much lower rate than that in patients with cirrhosis [46].

#### 3.5.3. Fulminant Hepatitis Model

The LEC rats (Long–Evans rats with cinnamon-like coat colors), an inbred strain separated from Long–Evans rats, develop hereditary hepatitis with symptoms of jaundice without hepatitis virus infection. The clinical signs of hepatitis resemble those of human fulminant hepatitis [47]. LEC rats suffer a fulminant hepatitis in which approximately 50% of rats die. All aged rats that survived for more than 1.5 years developed well-differentiated HCC.

LEC rats were characterized by excessive accumulation of copper and iron in the liver [47]. These transition metals produce ROS (hydroxyl radicals) in hepatocytes through the Fenton reaction [47]. Therefore, LEC rats serve an adequate model for ROS-induced HCC due to fulminant hepatitis. Moreover, hereditary hepatitis in LEC rat hepatocytes shows steatosis in the cytoplasm and pleomorphic mitochondria closely associated with copper toxicity, which may deal with a rat form of Wilson’s disease [48]. Deletion of the ATP7B gene, which is homologous to the Wilson’s disease gene, has been identified in LEC rats [49].

#### 3.5.4. Parasitic Infection Model

Three eukaryotic pathogens—namely, schistosomiasis, liver flukes (*Opisthorchis viverrini* and *Clonorchis sinensis*), and *Helicobacter pylori*—are considered Group 1 carcinogens by the International Agency for Research on Cancer (IARC) [9,50]. It has been generally accepted that cholangiocarcinoma in a hamster model occurs only as a result of a combined action of infection with a liver fluke, *Opisthorchis viverrini* (*O. viverrini*), *Clonorchis sinensis* (*C. sinensis*), and *Opisthorchis felineus* (*O. felineus*), together with exposure to carcinogens [50].

It has been epidemiologically confirmed that food-borne liver fluke, *O. viverrini*, infection causes intrahepatic cholangiocarcinoma in a geographically limited and endemic area of Southeast Asia [51]. Hamsters, jirds, guinea pigs, and rabbits are susceptible hosts for *O. viverrini*. A combination of intragastrical infection with *O. viverrini* and the administration of the sub-carcinogenic dose of the carcinogen dimethylnitrosamine has succeeded in a reproducible inflammation-related cholangiocarcinogenesis model in Syrian golden hamsters [52]. The pathological consequences of chronic *O. viverrini* infection occur primarily in the intrahepatic bile ducts, which then transitions into bile duct hyperplasia associated with either chronic cholangitis or pericholangitis where cholangiofibrosis and cholangiocarcinoma arise. Inflammatory cells comprise eosinophils, monocytes/macrophages, T cells, and plasma cells [50,51].

The liver fluke *O. felineus* is another potential parasite that causes inflammation-related cholangiocarcinogenesis. *O. felineus* infection occurs primarily within the territory of the Russian Federation, and it is now increasingly seen in other European regions [50]. Hamsters were intragastrically infected with *O. felineus*, the metacercariae exist in the duodenum and the juvenile parasite ascended through the ampulla of Vater into the bile ducts, where the adult worm develops. The gallbladder and extrahepatic bile duct infiltration of monocytes and eosinophils into the portal vein area in response to the parasitic infection cause adenomatous hyperplasia, epithelial hyperplasia, and granulomatous inflammation. It is then formed in the bile duct epithelium and identified as a biliary intraepithelial neoplasia caused without a carcinogen [9].

#### 3.5.5. Diet-Related and Obesity Model

##### Alcohol

Alcohol is a contributing risk factor to a variety of medical conditions, including cancers of the mouth, esophagus, pharynx, larynx, and liver. Given the free choice, most rodents do not voluntarily consume large amounts of alcohol. Therefore, previous studies have adapted a model in which rodents have “forced” consumption of alcohol [53].

In the forced alcohol intake model, the Lieber–DeCarli alcohol liquid diet is administered to the mice, and the alcohol concentration is gradually increased in a stepwise manner. Feeding the mouse with a 4% Lieber–DeCarli alcohol diet after injection of the carcinogen *N*-diethylnitrosamine (DEN) reduces steatosis, increases fibrosis, and leads to symptoms of advanced liver disease. In alcohol/DEN mice, both neutrophils and M2 macrophages infiltrate into the livers and cause the development of intrahepatic cysts and HCC [54].

A selective breeding program to increase the demand for ethanol led to a rat strain named “P rat” that voluntarily drank large amounts of alcohol [54]. The P rat model mimicked the level of alcohol consumption by humans who habitually abuse alcohol [53]. When given the free choice between a 10% alcohol solution and water, P rats voluntarily consumed 6–8 g alcohol/kg/day. That is nearly comparable to the alcohol intake of a man who drinks two six-packs of beer daily. P rats that received 10% alcohol continuously for more than one year developed HCC. The potential cause for the induction of HCC in P rats with alcohol consumption was that the relevance of hepatic oxidative status was increased. Intriguingly, like human alcoholics, alcohol-consuming P rats develop HCC in the absence of steatohepatitis, liver fibrosis, or cirrhosis. Moreover, alcohol ingestion was shown to enhance HCV-related tumorigenesis in the mouse models [53].

##### Obesity

Aberrant lipogenesis in the liver, which is closely linked to obesity and metabolic syndrome, causes nonalcoholic fatty liver disease (NAFLD), which is observed in 75–100% of overweight and obese adults and children [55]. Steatosis is the initial stage of NAFLD, which can progress into the more severe form of NAFLD called nonalcoholic steatohepatitis (NASH), fibrosis, and cirrhosis, with the result of an increased risk of HCC development [40]. NASH is associated with insulin resistance, increased oxidative stress, and induced inflammatory cytokines and can ultimately lead to cirrhosis and HCCs [55].

There was a model that spontaneously developed obesity, hyperlipidemia, diabetes mellitus, NAFLD/NASH, and liver carcinogenesis after a single injection of monosodium glutamate (MSG) in newborn mice. MSG induced beta cell proliferation and caused islet hypertrophy [56,57], leading to the development of obesity and steatosis with the infiltration of neutrophils and the onset of diabetes mellitus. NAFLD/NASH-like liver lesions and steatosis with the infiltration of inflammation were observed, which then developed into HCC. The latency period shortened but tumor incidence increased using a carcinogen, DEN [56].

A methionine-choline-deficient (MCD) diet without carcinogens induced NAFLD, and hepatocarcinogenesis was observed using inducible liver-specific myc oncogene transgenic mice [58,59]. In this model, a MCD diet caused the selective loss of intrahepatic CD4^+^ but not CD8^+^ T cells, which promoted the dysregulation of NAFLD lipid metabolism-mediated ROS generation and led to HCC. Among the patients with NASH, alcoholic steatohepatitis, and viral hepatitis, a specific loss of intrahepatic CD4^+^ was observed only in the NASH patients [58].

There was a rat model in which HCC develops in the hepatic NASH induced by the MCD diet and DEN treatment. Inflammation-related cytokines, especially IL-6 and ROS generation, were observed. Preneoplastic glutathione S-transferase placental form (GST-P)-positive foci formed in the livers at a very early phase [60].

A semisynthetic choline-deficient L-amino acid-defined (CDAA) diet does not contain any of the vitamin choline and is low in methionine, leading to HCC with a background of fatty liver and hepatocyte death and a subsequent regeneration and fibrosis, resulting in cirrhosis in rats without exposure to a carcinogen [61]. Interestingly, the model closely resembles the pathobiology of NASH, and there is no weight gain with the CDAA diet. This model can also be applied to NAFLD or NASH studies without obesity, mainly found in Asian regions.

Livers of high-fat diet (HFD)-fed mice showed typical obesity-induced chronic inflammation, steatosis, and NAFLD and NASH and, finally, HCC development; however, cirrhosis was not developed in mice. It is known that approximately half of human liver tumors have G–T (Q61K) mutations at codon 61 of the H-ras oncogene. Whole-exome sequencing of liver tumors formed in HFD mice showed that similar G–T point mutations occurred predominantly. Fatty liver diseases are thought to induce oxidative stress and lipid peroxidation, resulting in a typical gene signature of the G–T mutation [62]. Adding DEN to HFD feeding increases hepatic inflammation and the incidence of HCC and shortens the latency period [63]. Moreover, instead of HFD, a high-carbohydrate diet and DEN administration also stimulated hepatic inflammation and formed HCC in mice [40].

#### 3.5.6. Microbe Model

*Helicobacter marmotae* infection exerts a tumor-promoting activity in woodchuck livers to induce chronic inflammation, which, in turn, accelerates neoplastic conversion in hepatocytes [64].

#### 3.5.7. Chemical Carcinogenesis Model

DNA adduct-forming agents in environmental carcinogens are also important risk factors for HCC. DEN has been widely used as a potent hepatocarcinogenic initiator in rodent models in which it induces DNA adduct formation, resulting in DNA mutations. DEN preferentially generates well-differentiated HCC [65].

A rat two-stage liver carcinogenesis model with DEN initiation and thioacetamide (TAA) promotion has been established. TAA acts as a proinflammatory mediator and induces liver injury and fibrosis. The chronic administration of TAA typically causes recurrent hepatocyte injury, which is followed by the regeneration of hepatocytes, inducing oxidative stress and causing regenerative nodules, liver cirrhosis, and, eventually, forming HCC. TAA stimulates the infiltration of inflammatory cells, hepatic macrophages, and CD3^+^ T cells [66].

In mice, treatment with cholic acid, a typical bile acid, increases the number and size of DEN-initiated liver tumors with inflammatory cell infiltration, such as CD3, CD20, and CD45-positive cells [67].

Treatment with carbon tetrachloride (CCl_4_) induces liver fibrosis and is an accepted model to mimic human disease. The carcinogenicity and chronic toxicity of CCl_4_ were observed by inhalation exposure in rats for around two years. CCl_4_ induced chronic hepatotoxicity, such as fatty change, fibrosis, and cirrhosis, and led to the development of hepatocellular adenomas and HCC. A cytotoxic-proliferative and genotoxic mode of action for CCl_4_-induced hepatocarcinogenesis was suggested [68].

#### 3.5.8. Genetically Modified Host Model

Autophagy, an intracellular lysosomal degradation pathway, acts as a tumor suppressor. The hepatocyte-specific homozygous deletion of Atg5 (Atg5^−/−^) mice resulted in increased inflammation and fibrosis and led to the formation of hepatocellular adenomas in the liver that was infiltrated with neutrophils and macrophages [69].

The heterozygous loss of β2-spectrin in mice (β2SP^+/−^) led to the development of HCC due to the dysregulation of TGF-β signaling. Tumor cells obtained from inflamed liver tumors by introducing a prophlogistic IL-6-encoding DNA plasmid into β2SP^+/−^ mice were more malignant than those obtained from tumors developed in IL-6-introduced wild-type mice [70].

The farnesoid X receptor (FXR and NR1H4), a member of the nuclear receptor superfamily, controls the synthesis and transport of bile acids in the liver and gut. The homozygous deletion of the farnesoid X receptor (Fxr^−/−^) in mice results in the accumulation of high levels of bile acids, which provokes inflammation-induced hepatocarcinogenesis with fibrosis and hepatosteatosis [71].

Gankyrin was an oncoprotein that was overexpressed in human liver cancers. Hepatocyte-specific gankyrin-expressing transgenic mice have high serum levels of inflammatory cytokines (TNF-α and IL-6) and enhanced infiltration of macrophages into the liver. When the irritant CCl_4_ was administered to these mice, fibrosis was formed, and then, liver tumors developed [72].

Mice with homozygous mutated diabetes genes (db) (db/db or Lepr^db^/Lepr^db^) have a functional defect in the long-form leptin receptor, leading to hyperleptinemia and obesity due to overeating. Coexistent obesity or steatosis exacerbates liver injuries and fibrosis and, thus, is involved in liver tumorigenesis. The db/db mice received DEN and developed hepatic preneoplastic lesions, the foci of cellular alteration, adenoma, and HCC, accompanying the expression of inflammatory cytokines such as TNF-α, IL-1β, and IL-6 [73].

The mouse mdr2 gene encodes a P-glycoprotein present in the bile canalicular membrane of hepatocytes. Due to the insufficient secretion of phospholipids into the bile, mice with a homozygous deletion of the Mdr2 P-glycoprotein (Mdr2^−/−^) gene develop nonsuppurative inflammatory cholangitis, which mainly contributes to B cells and stimulates tube proliferation, and most mice develop HCC [74].

Activation-induced cytidine deaminase (AID) is a nucleotide-editing enzyme, and the aberrant expression of AID is induced by inflammation. Since tissue-nonspecific alkaline phosphatase (TNAP) is specifically expressed in mouse hepatocytes, double-transgenic (TNAP-AID) mice showed liver-specific AID expression. In TNAP-AID mice, AID is specifically expressed constitutively in hepatocytes, and these mice developed chronic hepatitis, cirrhosis, and HCC. HCCs generated in TNAP-AID mice acquired the same mutational signatures as those occurring in hotspots of the p53 gene observed in human liver cancer [75].

### 3.6. Pancreas Cancer

Chronic pancreatitis increases the risk of developing pancreatic ductal adenocarcinoma (PDAC), one of the deadliest human cancers, by 16-fold [76].

#### Genetically Modified Host Model

Oncogenic K-ras mutation represents the most frequent and earliest genetic alteration in PDAC patients, which highlights its role as a driver of PDAC [77]. Relevant mouse models of PDAC have been generated by targeting a conditional activation of mutated K-ras allele (K-ras^G12D^ or K-ras^G12V^) in a transgenic mouse the early pancreatic progenitors, which indicates that K-ras mutation is sufficient for PDAC initiation. The incidence of spontaneous pancreatic cancer is low in these mice observed after more than one year. However, inducing inflammation or by mating with other genetically modified mice, it has become possible to increase the carcinogenic frequency and shorten the latency period [76].

In K-ras^G12D^ mice, the administration of a pancreatitis-inducer, cerulein, causes acinar cell death and induces an inflammatory response. Cerulein treatment induced chronic pancreatitis with fibrosis, involving the infiltration of neutrophils, eosinophils, macrophages, T cells, and B cells. Cerulein treatment shortened the latency period and increased the incidence of pancreatic carcinogenicity [76,77]. In addition, a link between PDAC and the intake of a high-fat diet (HFD) has been shown in K-ras^G12D^ mice. That is, HFD activates oncogenic K-ras with COX-2 upregulation, which causes inflammation and fibrosis of the pancreas, as well as the development of pancreatic intraepithelial neoplasia (PanINs) and PDAC [78].

In K-ras^G12V^ mice, the epithelial transdifferentiation observed in acinar-to-ductal metaplasia occurs in the pancreas in response to inflammation. K-ras^G12V^ mice form preneoplastic PanINs, and its conversion recapitulated what was observed in human PDAC. After more than one year, mice develop pancreatic cancer very occasionally. However, the cooperative expression of the hemizygous p53 gene (K-ras^G12V^p53^+/-^) died before six months by poorly differentiated PDAC [76,77].

### 3.7. Colorectal Cancer

Chemical carcinogen-induced or genetically modified models of intestinal inflammation do not completely mimic the diseases status found in inflammatory bowel disease (IBD) patients [79] but are readily available and reproducible.

#### 3.7.1. Chemical Irritant Model

A nongenotoxic carcinogen, dextran sulfate sodium (DSS), is a synthetic sulfated polysaccharide composed of dextran with sulfated glucose. DSS induces colonic inflammation and its associated dysplasia and carcinomas in guinea pigs, rabbits, hamsters, and mice with clinical and histopathological similarity to human ulcerative colitis (UC) [80,81]. DSS causes inflammation mainly in the distal colon [80].

Inducing inflammation-related colorectal cancer by DSS alone requires long-term administration, and tumor incidence and multiplicity are low [79]. In hamsters, DSS induces inflammation and multiple erosion but only in the large intestine. Six months after continuous administration of DSS, the development of colorectal dysplasia, adenoma, and adenocarcinoma were evident [81]. In mice, the administration of DSS was repeated intermittently under conditions similar to those in active and remission clinically observed in UC patients, with colorectal dysplasia and carcinoma occurring at approximately seven months [80]. In rats, a short-term exposure to DSS results in the formation of aberrant crypt foci (ACF), a putative precursor lesions of adenocarcinoma at six weeks, while long-term exposure causes papilloma, adenoma, and adenocarcinoma until seven months [80].

By using a genetically modified host, colitis is expressed in a more severe form, and the incidence of carcinogenicity increases.

Thrombospondin 1 (TSP-1) is a multidomain glycoprotein involved in angiogenesis. TSP-1^−/−^ mice showed rectal bleeding, crypt injury, inflammation, increased dysplasia, and early onset of colorectal cancers by DSS without a carcinogen [82].

Carrageenan is a nongenotoxic sulfated polysaccharide that induces ulcerative colitis in the proximal or transverse colon in guinea pigs, rabbits, rats, and rhesus monkeys [83]. In rats, the long-term administration of carrageenan results in squamous metaplasia and adenomatous polyps of the rectum. Before epithelial conversion, carrageenan-induced inflammation is characterized by macrophage aggregation in the lamina propria and submucosa. The development of adenoma, squamous cell carcinoma, and adenocarcinoma is evident until two years [83].

#### 3.7.2. Chemical Carcinogenesis Model

The intestinal inflammation-related carcinogenesis model has a shortened latency period when induced using the carcinogen azoxymethane (AOM), which is a metabolite of 1,2-dimethylhydrazine. AOM combined with DSS showed a multiplicity of colonic cancers [78]. The severity of inflammation depends on the amount of DSS, as reduced doses of DSS formed dysplastic crypts rather than cancer [84].

The AOM/DSS model established by Tanaka et al. provided reproducible colorectal carcinogenesis accelerated by inflammation by selecting the mouse strains and AOM doses [80]. This model showed proinflammatory molecules (TNF-α, IL-1β, IL-6, COX-2, β-catenin, NF-κB, STAT3, inducible nitric oxide synthase (iNOS), matrix metalloproteinase (MMP)-7, and MMP-9) but did not show p53 immunoreactivity [76,80,82]. The inflammatory response of the AOM/DSS model is similar to that of human colitis in terms of the type of infiltrated inflammatory cells [80].

Mouse strain differences have been demonstrated to play a role in susceptibility to the model. The tumor incidence was 100% in Balb/c mice, 50% in C57BL/6 mice, and only a few adenomas occurred in C3H/HeN and DBA/2N mice [85]. Differences in the tumorigenic frequency reflect the ability of the mice to induce inflammation and sensitivity to DSS.

2,4,6-Trinitrobenzenesulfonic acid (TNBS) has tumor-promoter properties. The administration of TNBS to AOM-injected mice through a rectal catheter causes severe colonic inflammation, with the infiltration of neutrophils into the submucosal and muscle layers. Inflammation induced by chemical irritants may be affected by the gut microbiota, because antibiotic treatment and the use of germ-free animals reduce inflammation [86,87].

Psychosocial stress also increases inflammation. There is a chronic subordinate colony housing model in which a large dominant male mouse regarded as a chronic stressor and a few mice with small body size are housed together, causing post-traumatic stress disorder (PTSD) in smaller mice [88,89]. This model has been substituted for a clinically relevant model of psychosocial stress, as it causes the onset of IBD and exacerbates DSS-induced colitis. In this stressed mouse, there is a switch from tumor-protective Th1 to regulatory T cell immunity, representing increased inflammation and carcinogenesis [88,89,90].

#### 3.7.3. Diet-Related and Obesity Model

Obesity and its related metabolic abnormalities are associated with the development of colorectal cancer [91]. The adipose tissue serves as a natural reservoir of macrophages and inflammatory cytokines, especially IL-6, which suggests that obese people may always be exposed to chronic inflammation [92].

There are two approaches for establishing obesity models: One is the use of a high-lard-and-sucrose content diet named the “Western diet”, while the other one is the use of a diet with low amounts of calcium, vitamin D3, and other nutrient profiles that emulate a typical Western diet named the “stress diet”, using the available U.S. nutrient intake survey data [93].

The Western diet-induced obesity model includes a high-calorie diet (high-fat and high-sugar). The long-term administration of a Western diet to mice resulted in the induction of colon tumors. These obese mice had increased colon cell proliferation and induction of ACF and colon cancer compared to the lean mice after AOM injection. The Western diet affected the inflammatory responses, especially in the activating IL-6 signaling pathways [94].

The stress diet-induced obesity model includes a reduced dietary intake, which has been suggested as a risk factor for human colon cancer. The stress diet increased the fat content of mice and resulted in the development of colonic hyperplasia, adenoma, and carcinoma without exposure to the carcinogen. Notably, stress diet-fed AOM-initiated or Apc^Min/+^ mice had severe, prolonged colitis and a high incidence of tumor in DSS-induced inflammation [94,95].

#### 3.7.4. Genetically Modified Host Model

The multiple intestinal neoplasia (Min) mouse, which carries a germline mutation-converting codon 850 of the murine Apc gene, which converts leucine to a stop codon, was established by phenotypic screening after a random germline mutagenesis with ethyl-nitrosourea, a carcinogen. The Min/Min homozygous mutation of the adenomatous polyposis coli gene in mice is lethal, and mice with a heterozygous mutation (Apc^Min/+^) survive but develop adenomas throughout the small intestine and only rarely in the colon; however, in humans with hereditary familial adenomatous polyposis, adenomas occur in the colon, duodenum, and rectum [96,97].

Exposure of DSS to a Apc^Min/+^ mouse accelerates colitis, colonic dysplasia, and cancer. The findings suggest that DSS-induced inflammation in the large bowel of Apc^Min/+^ mice has a tumor-promotive effect on the growth of dysplastic crypts that were already present after birth [98].

There are three mouse strains with different codon sites for APC mutations, and the difference in mutation is represented by the difference in tumorigenesis site and number. Apc^+/Δ716^ mice are similar to APC^Min/+^ mice and develop a large number of adenomas in the upper gastrointestinal tract. Apc^+/1638N^ mice develop tumors in regions spanning from the stomach to the small intestine of the upper gastrointestinal tract, which is a phenotype different from that of the Min model. Apc^+/1638N^ mice form gastric tumors, osteomas, desmoid tumors, and mammary carcinoma. Apc^+/1638T^ mice have a similar codon mutation as that of Apc^+/1638N^ mice but do not develop tumors [99].

Since colorectal carcinogenesis progresses through multiple stages of genetic alteration, Apc^Min/+^ mice limit a whole understanding of carcinogenesis due to a single gene-deficient model [97]. Therefore, researchers mated Apc^Min/+^ mice with other gene-deficient or knock-in mice to produce offspring with multiple gene alterations. The co-expression of proinflammatory mediators such as IκB kinase-β (IKK-β), IL-6, IL-8, and IL-17A in Apc^Min/+^ mice causes severe inflammation and accelerates intestinal tumorigenesis [99]. Intestinal tumorigenesis is also accelerated by mating with mice that have lost their inflammation-suppressive molecules. All offspring mice were mated Apc^Min/+^ mice with homozygous or heterozygous knockout mice, and the following inflammatory molecules showed increased carcinogenesis through the induction of severe inflammation. The regulator molecules included Fas/CD95/Apo-1 receptor, Rag1 or Rag2, GST-P, Nrf-2, Tpl2, TRLs, IL-1R, Sigirr, IFN-γ, or IFN-γR1 [97,100,101].

IL-10 is an anti-inflammatory cytokine. The homozygous deletion of interleukin-10 (IL-10^−/−^) mice leads to enterocolitis, which is similar to human IBD. IL-10^−/−^ mice develop progressive inflammatory changes in the colon and colorectal adenocarcinoma. The treatment of IL-10^−/−^ mice with AOM/DSS caused colitis and tumors as a result of microbial activation, but tumor development was suppressed under germ-free conditions. Moreover, Apc^Min/+^IL-10^−/−^ mice developed colon tumors, whereas this was also suppressed under germ-free conditions. The effects of intestinal microbiota on the development of enteritis in IL-10^−/−^ mice have become apparent [43,102].

KK mice are a model of type 2 diabetes with intact leptin and leptin receptors. C57BL/6-Ay mice show severe hyperphagia, polydipsia, impaired glucose tolerance, hyperinsulinemia, and hyperlipidemia. The KK-A^y^ mice established by cross-mating KK mice with C57BL/6-Ay mice developed ACF and tumors following the increase in infiltrated macrophages [92,103].

Since the diabetes (db) gene encodes the receptor for the obese (ob) gene product, leptin, mutations in the mouse db gene, cause leptin dysfunction, resulting in obesity and diabetes in a syndrome similar to that of human obesity. Leptin induces proinflammatory cytokines (TNF-α and IL-6), which leads to inflammation. The homozygous mutated diabetes (db/db) mice are susceptible to AOM and develop ACF and β-catenin-accumulated crypts [89].

Lactoferrin (LF) exerts anti-inflammatory and antitumor activities. The homozygous deletion of lactoferrin (Lf^−/−^) mice showed higher susceptibility to AOM/DSS-induced colitis and developed dysplasia [104].

Nuclear factor, erythroid-derived 2-related factor 2 (Nrf2), is a transcriptional regulator of multipole cytoprotective genes that contribute to the cellular redox balance, immunity/inflammation, metabolism, and carcinogenesis. The homozygous deletion of Nrf2 (Nrf2^−/−^) mice treated with AOM/DSS demonstrates increased colitis and adenocarcinomas. In these mice, higher levels of inflammatory cytokine mRNAs and neutrophil infiltration within the colon were observed [101,105].

Syndecan-1 (Sdc1) is a cell surface heparan sulfate proteoglycan expressed by epithelial cells. The homozygous deletion of syndecan-1 (Sdc1^−/−^) mice results in an increased susceptibility to colitis-associated cancer induced by AOM/DSS through IL-6 and STAT3 signaling [106].

#### 3.7.5. Microbe Model

*Salmonella*, a Gram-negative bacterium, causes gastroenteritis and sepsis, as well as intestinal injury in both humans and animals. *Salmonella* infections increase the risk of IBD. The infection of transgenic mice overexpressing c-myc in Apc^+/min^ with *Salmonella typhimurium* caused in colitis and cancer but not when infected with mutant *Salmonella* (bacterial proteins cannot be transferred into the host cytosol) infection [107].

#### 3.7.6. Foreign Body-Induced Inflammation Model

A mouse model in which inflammation caused by foreign body substance promotes carcinogenesis of human-derived cells has been established. A culture cell line of adenoma cells (FPCK-1-1 cells) was originally established from a colonic polyp of a patient with familial adenomatous polyposis. The phenotype of the cells was stable, and there was no spontaneous tumorigenic conversion during regular cultivation. FPCK-1-1 cells did not grow in nude mice, but when attached to a plastic plate and implanted into subcutaneous space of mice, they formed moderately differentiated adenocarcinoma surrounded by highly fibrous stroma in six months. The fibrous tissue, rather than attachment to the plastic plate substrate, was considered essential for the carcinogenesis, because FPCK-1-1 cells were converted into tumorigenic ones by injecting the cells directly at the site of proliferating stromal tissues where the plastic plate had been implanted for six months and then removed. The inflammation essential for this carcinogenesis was chronic rather than acute, since co-implantation with the acute inflammation-inducing substance, gelatin sponge, did not cause tumors [108]. Increased expression of the actin-related cytoskeletal protein fascin and miR-146a, which regulates the stability of fascin, has been identified to cause genetic alterations associated with this model [109,110].

### 3.8. Breast Cancer

Monocytes and macrophages are central to inflammation-related breast carcinogenesis. Monocyte chemoattractant protein-1 (MCP-1) and tumor-associated macrophages are prognostic indicators of both relapse-free survival and overall survival in primary breast tumors [111].

#### 3.8.1. Diet-Related and Obesity Model

Dietary lipids are important factors influencing the breast cancer etiology. Their effect on mammary carcinogenesis depends on, among other factors, the amount and type of fat and the timing of dietary interventions. Rats were fed a high-fat diet, i.e., a corn oil-based or an olive oil-based diet, after DMBA initiation. The corn oil-based diet had a clear tumor-promoting effect on breast carcinogenesis, while the olive oil-based diet had a suppressive effect. This is due to the presence of different types of infiltrating leukocytes. The corn oil-based diet showed an increase in arginase expression and IL-1α, while the olive oil-based diet increased CD8^+^ but not CD4^+^ T cells and reduced TGF-β1 expression [112].

#### 3.8.2. Foreign Body-Induced Inflammation Model

Using weakly tumorigenic and nonmetastatic clonal ER-1 cells by exposing the SST-2 culture cell line, which is established from a mammary adenocarcinoma developed spontaneously in SHR rats, to ethyl-methanesulfonate in vitro, ER-1 cells regressed after injection into syngeneic normal rats; however, if they were implanted, as attached to plastic plates, into a subcutaneous space in rats, they acquired not only stable tumorigenic phenotypes but also metastatic ability [113]. The progression of ER-1 cells requires fibrous stroma and the associated epidermal growth factor (EGF) or TGF-ß stimulation. ER-1 cells were converted into tumorigenic ones by continuously adding EGF into the culture. The acquisition of the tumorigenic phenotype depended on the duration of the EGF treatment; the tumorigenic phenotypic was reversible during the 24 h exposure to EGF, and after more than four weeks of exposure, the cells acquired irreversible phenotypes. This is why chronic inflammation is required for breast carcinogenesis, because the proper concentration and duration of repeated treatments are important [114]. The addition of antioxidants to EGF treatment prevents the formation of ROS-mediated nucleic acid damage and the acquisition of tumorigenic phenotypes [115].

#### 3.8.3. Genetically Modified Host Model

The C3 (1) component of the rat prostate steroid-binding protein is used to target the expression of the SV40 large T-antigen (Tag) to the epithelium of both the mammary and prostate glands; this results in the models of mammary and prostate cancers. In the C3(1)/SV40Tag transgenic mouse, the recruitment and activation of tumor-associated macrophages based on the expression of MCP-1 are required for the development of breast cancers [111].

There are two different transgenic mouse models of breast carcinogenesis that carry activated tyrosine kinases.

First, the activated tyrosine kinase was neu. This spontaneous mammary carcinogenesis model was established in transgenic mice carrying the activated neu oncogene (^V^664^E^) driven by the murine mammary tumor virus long terminal repeat (MMTV-LTR) [116]. In this mouse, the protein product of the neu oncogene (p185^neu^) is overexpressed in the terminal buds of mammary glands. The infiltration of immunosuppressive T_reg_ cells and neutrophils deleted CD8^+^ T cells that recognize p185^neu^ epitopes and formed mammary tumors [117]. In addition, the infiltrated Tie-2-expressing monocyte and its derived TNF-alpha are involved in the development of mammary carcinogenesis [118].

The second is another potent tyrosine kinase activity driven by the polyomavirus middle T antigen (PyMT). Transgenic mice expressing MMTV-PyMT spontaneously developed mammary adenocarcinoma [119]. In the MMTV-PyMT mouse model, mammary tumor growth induces the accumulation of tumor-associated macrophages (TAMs), which are phenotypically and functionally distinct from mammary tissue macrophages. TAMs differentiated from inflammatory monocytes specifically inhibit tumor-infiltrating cytotoxic T cells, causing local immunosuppression and promoting mammary carcinogenesis [120].

### 3.9. Urogenital Cancer

#### 3.9.1. Prostate Carcinogenesis Model

ACI/Seg (ACI) rats spontaneously develop invasive prostate cancers during aging, similar to humans. Comparing gene expression patterns in the prostates of young and old ACI rats, changes in the genes associated with inflammation, oxidative stress, and tissue remodeling were observed in older rats. Hyperplasia and atrophy were observed in the precancerous lesions of the prostate in ACI rats. These lines of evidence indicate that the involvement of chronic inflammation and the associated oxidative stress accelerate prostate carcinogenesis [121,122,123].

The long-term administration of estradiol-17β or diethylstilbestrol to rats, in addition to testosterone, increases the incidence of prostate adenocarcinoma. NBL rats are sensitive to the development of hormone-induced prostate carcinoma compared to Sprague–Dawley rats. All rats had focal dysplastic and inflammatory lesions in the prostate and pituitary organs and formed prostate adenocarcinomas and pituitary adenomas. Sex-hormones induced prostatic inflammation and oxidative stress in rats, which resulted in multifocal adenocarcinomas originating from the ducts of the prostate [123].

Benign prostatic hyperplasia (BPH), which is associated with the development of prostate cancer, affects men over the age of 50, and its prevalence increases with age. One BPH model is based on hyperplasia of the prostatic lobes by administering prolactin to male rats, which is also an inflammation-related prostate carcinogenesis. The model shows leukocyte infiltration into the prostate tissue and develops typical nonbacterial prostatitis [124].

To determine the role of chronic inflammation in the carcinogenesis using prostate epithelium, a prostate tissue regeneration system was developed. Prostate epithelial tissue is obtained from PTEN-conditionally knockout mice, and after the overexpression of inflammatory cytokines (IL-6 or oncostatin-M), the tissue grafts were transplanted under the renal capsule of mice to confirm acquisition of tumorigenicity. Grafts expressing inflammatory cytokines showed high-grade prostatic intraepithelial neoplasia and, subsequently, progressed to poorly differentiated adenocarcinoma. The same results were also observed by using benign human prostate epithelium [125].

#### 3.9.2. Bladder Carcinogenesis Model

Urogenital schistosomiasis, which is primarily caused by *Schistosoma haematobium* (*S. haematobium*) and its deposited eggs, triggers chronic inflammation. Urogenital schistosomiasis, which occurs in the bladder and genital tract, results in an increased risk of carcinogenesis. Animal models for urogenital schistosomiasis were developed by administering an injection of *S. haematobium* eggs directly into the bladder walls of mice. This model recapitulates multiple aspects of human urogenital schistosomiasis, including immunological type 2 shifts, bladder granulomatous inflammation, and then the formation of urothelial hyperplasia [126].

#### 3.9.3. Renal Carcinogenesis Model

Renal cell carcinoma was developed by the repeated administration of ferric nitrilotriacetate (Fe-NTA). Fe-NTA induced severe acute nephrotoxicity and the infiltration of leukocytes into the proximal tubular epithelium, as well as the resulting oxidative stress and hyperproliferative response [127].

Renal mesenchymal tumors developed in approximately two years after administering an intrarenal injection of nickel sulfide into the kidney. Nickel induced extensive necrosis, inflammation, fibrosis, and degenerative and regenerative proliferative changes in the proximal tubular epithelium at the injection site. A mixed injection of magnesium carbonate or iron powder resulted in the development of renal tumors with more severe and multiple inflammatory changes [128].

### 3.10. Skin Cancer

Major risk factors for developing non-melanoma skin cancer include occupational and environmental exposure to polycyclic aromatic hydrocarbons and solar ultraviolet B radiation [129]. There are models specific for these two factors.

#### 3.10.1. Classical Two-Stage Skin Carcinogenesis Model

The classic two-stage carcinogenesis, which includes tumor initiation and promotion, remains conceptually important for skin carcinogenesis research. A typical two-step model uses DMBA as a common polycyclic aromatic hydrocarbon as an initiating agent, and prophlogistic 12-O-tetradecanoylphorbol-13-acetate (TPA) is used as a tumor-promoting agent. This model has been utilized to mimic the skin carcinogenesis of human squamous cell carcinoma. In the initiation phase, a low dose of carcinogens was used to cause DNA damage, but it does not give rise to tumors over the lifespan of the mouse unless a tumor promoter is repeatedly applied. The promotion phase is characterized by increased inflammation, tissue remodeling, and hyperproliferation, all of which stimulate cell growth. The biological function of the promoter is primarily prophlogistic and can be summarized as an activation of the arachidonic acid cascade and induction of inflammatory cytokines [96,130,131].

#### 3.10.2. Ultraviolet B Exposure Model

Representative ultraviolet B (UVB)-induced skin carcinogenesis was established in SKH-1 hairless mice. UVB stimulates the release of IL-1α from keratinocytes, which are the major cell type of the epidermis, and induces the expression of cytosolic phospholipase A2, which is involved in arachidonic acid metabolism. These events are so-called cytokine and eicosanoid networks that maintain the inflammatory response and promote skin carcinogenesis [104,132].

#### 3.10.3. Genetically Modified Host Model

In response to the DMBA-initiated and TPA-promoted skin carcinogenesis protocols, benign papillomas occur much faster in heterozygous p53 gene deletion (p53^+/−^) mice compared to wild-type mice. Homozygous-deleted (p53^−/−^) mice have the same frequency of papillomas as that of wild-type mice, but papillomas progress to tumors at a much faster rate [96].

UVB induces signature mutations characterized by C–T or CC–TT mutations at pyrimidine–pyrimidine sequences. These mutations are detectable in several tumor-suppressor genes, particularly Patched 1 (Ptch1) and p53 in UVB-exposed skin sites. Since the homozygous deletion of Ptch1 is embryonic, Ptch1^+/−^ mice were used for mating with SKH-1 hairless mice. Ptch1^+/−^ SKH-1 mice develop skin cancer due to UVB- or ionizing radiation-induced inflammation, the characteristics of which are similar to those of typical sporadic human skin tumors. In Ptch1^+/−^SKH-1mice, the infiltration of neutrophils, macrophages, and myeloid-derived suppressor cells are evident [129,133].

Xeroderma pigmentosum (XP) is an autosomal recessive disorder characterized by skin cancer on sun-exposed areas. XP involves defective nucleotide-excision repair, and of the eight different groups (groups A–G and a variant), XP complementation group A is the most representative example of human being susceptible to skin cancer. That is, early-onset and multiple skin tumors developed when exposed to sunlight. Homozygous deletions in XP group A (XPA^−/−^) mice are sensitive to UVB and induced a severe inflammatory response. UVB-irradiated XPA^−/−^ mice developed dermal edema, primarily with infiltration of the neutrophils, and ulceration, followed by squamous cell carcinoma [134].

### 3.11. Sarcoma

#### 3.11.1. Chemical Irritant Model

Carrageenin, which causes acute inflammation induced by sulphated polygalactose (native undegraded carrageenan), involves the accumulation of macrophages with phagocytosed carrageenin, neutrophils, and fibroblasts. In such a persistent inflammatory environment, rats developed sarcomas after 1.5 years [81].

#### 3.11.2. Foreign Body-Induced Inflammation Model

There are unique models in which nontumorigenic cells obtained from benign tumors or premalignant tumors are converted into tumorigenic ones by foreign body-induced inflammation. Regressive clonal QR cells were established by the in vitro exposure of clonal tumorigenic fibrosarcoma cells to a mutagen. QR cells did not form tumors after a subcutaneous or intravenous injection into normal syngeneic mice [135]. However, the implantation of QR cells attached to a foreign body, plastic plate, or by injection into the preinserted gelatin sponge in mice induced lethal sarcomas [136,137]. It has been clarified that the inflammation-accelerated tumorigenesis of QR-32 cells progresses in a multistep manner and with stepwise gene alterations [3,138,139].

Two foreign bodies were used, the difference being in their ability to induce inflammation. One was a piece of plastic plate, which initially induces acute inflammation and then transitions to chronic inflammation, whereas the other was a piece of gelatin sponge, which induces acute inflammation, since it is naturally absorbed after implantation; therefore, such a transition from acute to chronic inflammation is unlikely to occur with the use of a sponge. By using these two types of foreign body substrates, the quality and duration of the inflammation are modulated, and it has become possible to quantitatively evaluate the inflammatory reaction and to determine the real nature of inflammation that promotes carcinogenesis from various angles [3]. The inflammatory cells collected from the subcutaneously inserted sponge can convert QR cells into tumorigenic ones if both cells are mixed and injected into mice [137]. Therefore, infiltrated inflammatory cells appear to be directly involved in the carcinogenesis.

The type of inflammatory cells required for the malignant conversion of QR cells in the acute inflammatory response was neutrophils. To prove the role of neutrophils in tumorigenesis and metastasis, neutrophils were eliminated by administering an antineutrophil antibody. All the tumors in the mice, either nontreated or treated with control rat IgG, acquired malignant phenotypes. In contrast, anti-neutrophil antibody-administered mice did not acquire malignant phenotypes [140]. Inflammation-associated sarcoma development was abolished in the integrin-beta-2 knockout mice, which lack neutrophil migration into an inflammatory site, since integrin-beta-2 is the key adhesion molecule for inflammatory cell exudation. Moreover, sarcoma development is inhibited in gp91^phox−/−^ mice, in which superoxide anions is not generated due to the lack of the major NADPH oxidase, gp91^phox^ [141]. Therefore, the infiltrated neutrophils and their associated ROS contribute to inflammation-related sarcoma development and progression.

#### 3.11.3. Genetically Modified Host Model

Kaposi’s sarcoma is caused by Kaposi’s sarcoma herpesvirus (KSHV) infection. KSHV infection induces spindle cell transformation and angiogenesis characteristic of Kaposi’s sarcoma through the expression of a viral G protein-coupled receptor (vGPCR). The KSHV vGPCR induces small GTPase Rac1, an inflammatory signaling mediator, in tumor development and in angiogenesis [132]. Constitutively active Rac1 (V12 mutant or RacCA) transgenic mice are sufficient to cause Kaposi’s sarcoma [142].

Spontaneous sarcoma development by inflammation induced by foreign body implantation in mice with heterozygous deletion of p53 (p53^+/−^) is associated with a higher rate of incidence and shorter tumor latency than those in wild-type mice. While 10% of p53^+/−^ mice had formations of subcutaneous sarcomas, 80% of p53^+/−^ mice developed sarcomas in the inserted foreign body. Most of the sarcomas developed by foreign body-induced chronic inflammation lost the remaining wild-type p53 allele, which means a complete loss of p53 function, through inflammation-mediated RNS. That is, the tumor-suppressor gene p53 is one of the drivers responsible for foreign body-induced sarcoma [143].

Inflammation-related carcinogenesis models of each organ using laboratory animals were organized by the cause of inflammation—that is, chemical or irritant-induced inflammation (Table 1), surgery, hormone or diet-induced inflammation (Table 2), genetically modified host-induced inflammation (Table 3), infection-induced inflammation (Table 4), and foreign body-induced inflammation (Table 5).

## 4. Clinical Aspects of Inflammation-Related Carcinogenesis

The information what we have learned from the organ-specific animal models of inflammation-related carcinogenesis from the past to the present has been able to verify that the persistent infiltration of inflammatory cells is an essential cause common to these models. To prevent or treat inflammation-related carcinogenesis in humans, it is effective to suppress the infiltration of inflammatory cells that cause the condition. We summarized an application-oriented method for evaluating the degree of inflammation and a specific methodology for suppressing inflammatory exudation.

### 4.1. Evaluation of Systemic Inflammation

Two categories of scores have been proposed to monitor the systemic inflammatory response, those derived from protein measurement and those based on counting inflammatory cells.

A method for measuring two proteins (C-reactive protein and albumin) called the modified Glasgow prognosis score (mGPS) has been developed as an index for evaluating the inflammatory state. mGPS has been validated in more than 60 studies (>30,000 patients) in 13 countries and provides an independent prognostic value for different cancer types and cancer patient status [171].

C-reactive protein (CRP) is a classical acute-phase reactant protein and was named so because of its high binding affinity to C-polysaccharide of *Streptococcus pneumoniae* [172]. CRP is a sensitive and widely used marker of systemic inflammation synthesized by hepatocytes in response to inflammatory cytokines, whose half-life is short because it is catabolized by hepatocytes in 19 hours [173,174]. Notably, elevated levels of CRP are associated with an overall risk of cancer and the risk of various organ cancers. The overall estimates have shown a 11% increase in the risk of all cancers for increased natural logarithm levels of CRP [174].

Inflammatory cell counts are the most accurate indicator of inflammatory status. Currently, the number of blood cells is used as an index, but in the future, if the number of exudative inflammatory cells at the inflammation site can be counted, then it is considered to be the best index for inflammation. Neutrophils, lymphocytes, monocytes, and platelets are the major cellular components of systemic inflammation. Counts of these cells, either singly or in combination, have a prognostic value in patients with a variety of common solid tumors. It includes the neutrophil-to-lymphocyte ratio (NLR), platelet-to-lymphocyte ratio (PLR), and lymphocyte-to-monocyte ratio (LMR). Elevated NLR and/or PLR and decreased LMR appear to be associated with decreased survival, but numerous cutoff values have been reported, which raises questions about the reproducibility and reliability of these markers. Moreover, it should be noted that leukocyte counts can be affected by geographical and ethnic differences—for example, the proportion of patients who had NLR ≥3 was 71% in the United States and Israel, 42% in China, and 30% in Italy [171].

### 4.2. Removing Inflammatory Cells from Systemic Circulation (Therapeutic Apheresis)

If the cause of inflammatory carcinogenesis is the exudation of inflammatory cells, reducing the number of inflammatory cells in the peripheral blood may help in the prevention of carcinogenesis. Currently, there are several ways to suppress the inflammatory response in the extracorporeal environment, and these therapies have already been clinically applied [175].

#### 4.2.1. Leukocytapheresis

There are two basic methods for the extracorporeal removal of leukocytes (leukocytapheresis), i.e., centrifugal and adsorptive methods. In the centrifugal method, blood cells are separated based on their gravity, and both leukocytes and plasma, which contain inflammatory cytokines, are removed. In the adsorptive method, the patient’s blood is collected and passed through a column or a filter that uses selective adsorption. Leukocyte and cytokines are removed, and the treated blood is then returned to the circulatory system. Leukocytapheresis is now a nonpharmacological therapeutic approach for mainly autoimmune-related disorders, including active UC [176].

#### 4.2.2. Photopheresis

In extracorporeal photopheresis, a buffy coat (apheresis) obtained from circulating blood is treated with a photosensitizer and then irradiated with ultraviolet A (UVA) and, finally, reinjected into the body. The precise mechanism remains unknown, as most leukocytes undergo apoptosis, but the monocyte fraction survives, because it is resistant to UV light, and when it returns to the body, it differentiates into dendritic cells and reregulates the immune response. Particularly, it suppresses the autoimmune reactions by increasing the number of functional regulatory T cells. Photopheresis treatment in patients with Crohn’s disease has significant effects, because it results in a reduction or discontinuation of steroids and maintenance of long-term remission [177].

## 5. Conclusions and Future Directions

This review reaffirmed that, in an organ-specific animal model of inflammation-related carcinogenesis, the pathogens that cause different types of carcinogenesis are clearly irrelevant, and the essential cause is persistent inflammation.

We noticed that not all inflammatory and dysregulated immunities lead to carcinogenesis; some of the inflammatory diseases are clearly unrelated to the risk of cancer. For instance, rheumatoid arthritis is not linked to cancer risk, even though the inflammatory regions in patients with rheumatoid arthritis show mutations in tumor-suppressor genes at similar frequencies to those of genes in the digestive tract tumors arising from chronic inflammatory reactions. Another point is that parasites (helminth, *Trichuris suis*) could diminish the inflammatory response [3]. In addition, patients with autoimmune hepatitis, a disease in which severe liver inflammation persists indefinitely, rarely develop HCC, unlike patients with hepatitis, due to the virus [45]. Clarifying these differences, pro-carcinogenic or anti/unrelated inflammation is an important consideration in determining which type of inflammation is essential for inflammation-related carcinogenesis. Intriguingly, it is experimentally shown that inflammatory carcinogenesis can also be affected by psychological factors [88,89,90]. Stressed and depressed patients overall had increased leukocytosis, high concentrations of circulating neutrophils, and reduced lymphocyte proliferation [178]. It is important to understand that the etiology of inflammation-related carcinogenesis includes infectious pathogens, foreign bodies, and the psychological state of the individual.

Treatment for return of the activated inflammatory response to normal levels remains necessary to achieve the prevention and treatment of inflammatory-related carcinogenesis. What is needed in the future is the expansion of treatments such as specific leukocyte depletion therapy that physically removes the inflammatory reaction in the body. However, considering the current effects of anti-inflammatory drugs and the limitations of their use, we should continue to strive to develop safer and more stable anti-inflammatory drugs with fewer side effects.

## Figures and Tables

**Table 1 cancers-13-00921-t001:** Chemical or irritant-induced inflammation-related carcinogenesis models.

Host/Strain	Genetic Manipulation	Carcinogen	Irritant or Manipulation	Incidence and Duration	Inflammatory Reaction	Reference
Swiss H mouse	No	No	Mainstream cigarette smoke	Lung adenosquamous carcinomas (52%), bronchioloalveolar carcinoma (20%), squamous cell carcinoma (18%), and squamous cell carcinoma in situ (5%) at 30 weeks	Mainstream cigarette smokes evoke chronic inflammation in the lungs	[20]
C57BL/6 mouse	No	B[a]p	LPS	Lung tumors (97%) at 30 weeks	↑ IL-1β, cleaved IL-1β, IL-18, NLRP3, NLRP6, and caspase-1	[23]
A/J mouse	No	NNK	LPS	Lung tumors (100%) at 27 weeks	LPS induces recruitment of macrophages and located at the periphery of the tumors ↑ NF-kB	[16]
C57BL/6 mouse	No	B[a]p	LPS	Non-small cell lung cancers (almost 100%) at 30 weeks	Acceleration of tumor malignancy by LPS-induced inflammation	[24]
BALB mouse	No	MCA	Butylated hydroxytoluene	Lung tumor (100%) at 20 weeks	Butylated hydroxytoluene-induced inflammation	[25]
WBN/Kob rat	No	No	Alloxan	Forestomach well-differentiated squamous cell carcinoma (20%) at 50 weeks	Accumulation of neutrophils and stimulation of superoxide production ↑ iNOS and COX-2	[35]
CD-1 mouse neonatal	No	No	Monosodium glutamate	HCC (43%) at 54 weeks	Infiltration of neutrophils, lymphocytes, and fibrosis	[57]
CD-1 mouse neonatal	No	DEN	Monosodium glutamate	Foci of cellular alteration (100%), adenoma (80%), and HCC (50%) at 21 weeks	Infiltration of macrophages and ROS ↑ TNF-α, IL-1ß, IL-6, F4/80 and CCL2	[56]
F344 rat	No	DEN	Thioacetamine	Pre-neoplastic hepatocellular lesions and glutathione S-transferase placental (GST-P) form at 8 weeks	Induction of hepatic macrophages and CD3^+^ T cells ↑ IL-1ß, COX-2, HO-1, and MMP-12	[66]
C57BL/6 mouse	Gankyrin^hep^ TG	No	CCl_4_	Hepatic tumors at 16 weeks	Chronic inflammation and fibrosis ↑ TGF-β, Collagen-1, and TIMP1	[72]
F344 rat	No	No	CCl_4_	HCC (6%) and hepatic altered cell foci at 104 weeks	Cirrhosis, fibrosis, and fatty changes	[68]
FVB mouse	K-ras^G12D^	No	Caerulein	Pancreatic cancer develop in K-ras^G12D^ mice at 48 weeks. Caerulein shortened to 11 weeks. Caerulein formed high-grade pre-malignant PanIN at 4 weeks, and low-grade PanIN at 8 weeks and developed PDAC	There were foci of cancer cells that were surrounded by a reactive stroma characterized by smooth muscle actin expression	[76]
C57BL/6 mouse	K-ras^G12V^	No	Caerulein	Pancreatic cancer never developed in K-ras^G12V^ mice at least until 80 weeks. Caerulein developed PDAC (100%) at 32 weeks	Caerulein induces chronic pancreatitis and fibrosis. Inflammatory infiltrates consisting of neutrophils and eosinophils, CD3^+^ T cells and B cells, and macrophages	[144]
Sprague-Dawley rat	No	No	Carrageenan	Colorectal squamous cell carcinoma (20%), adenocarcinoma (13%), adenoma (5%), and metaplasia (98%) at 96 weeks	Induction of colitis	[145]
Wister Furth rat	No	DMBA	No	Mammary carcinoma resistant strain	↑CX3CL1, IL11RA, IL-4, C3, CCL11, ITGB2, CXCL12, and CXCR7 ↓ CCL20	[146]
Sprague-Dawley rat	No	No	Carrageenan	Rectal squamous metaplasia and colonic adenomatous polyps at 24 weeks	Acute and chronic inflammatory, in which aggregation of macrophages in the lamina propria and submucosa	[147]
Wister rat	No	No	Carrageenin	Sarcomas (15%) at 93 weeks and 28% at 118 weeks	Neutrophils and macrophages with phagocytosed carrageenin, fibroblasts and mature collagen fibers were formed	[148]
Syrian hamster	No	No	DSS	Colorectal adenocarcinoma (50%), adenoma (13%), and dysplasia (50%) at 26 weeks	Infiltration of neutrophils	[81]
CBA mouse	No	No	DSS	Colorectal adenocarcinoma (8%) and dysplasia (52%) at 27 weeks	Focal inflammatory cell infiltration including neutrophils and gland loss or crypt abscess formation; mucosal ulceration was observed by DSS treatment	[149]
ACI rat	No	No	DSS	Colorectal papilloma (53%), adenoma (3%), squamous cell carcinoma (13%), adenocarcinoma (17%), cecum adenocarcinoma (3%), and intestinal tumor (73%) formed until 94 weeks		[150]
ACI rat	No	No	DSS	Colon adenoma (19%), adenocarcinoma (19%), papilloma (12%), cecum adenoma (23%), adenocarcinoma (8%), and intestinal tumor (58%) until 31 weeks	Infiltration of inflammatory cells and hemorrhage	[151]
C57BL/6 mouse	TSP-1^−/−^	No	DSS	Colonic dysplasia (66%) at 12 weeks	No significant changes in inflammation were observed between the genotypes; myeloperoxidase (MPO) activity was higher in the colon of TSP-1^−/−^ mice ↑ VEGF and bFGF	[82]
CD-1 mouse	No	DMH	Chrysazin	Colonic adenocarcinoma (48%) and adenoma (57%) at 54 weeks	Mucosal gland was hypertrophic and infiltration of inflammatory cells with fibrosis in submucosal areas	[152]
C57BL/6 mouse	KK-A^y^	AOM	No	Aberrant crypt foci (ACF) (100%) and colorectal tumor including well- and moderate-differentiated adenocarcinoma, and mucinous carcinoma (87%) at 25 weeks	Macrophage infiltration ↑ CSF1, IL-1ß, MCP1, and OPN	[103]
CBA mouse	No	AOM	DSS	Colorectal tumors at 11 weeks	Infiltration of inflammatory cells on the left side of the large intestine followed by the transverse colon	[83]
CD-1 mouse	No	AOM	DSS	Colonic adenocarcinoma (100%) and adenoma (38%) at 20 weeks	β-catenin-, COX2- and iNOS-positive inflammatory cells were infiltrated into mucosal ulceration	[80]
F344 rat	No	AOM	DSS	Aberrant crypts foci at 6 weeks	-	[153]
C57BL/6 mouse	No	AOM	DSS Chronic subordinate colony housing	Colonic dysplasia (100%) at 27 weeks	Increase regulatory CD3^+^FoxP3^+^ cells within mesenteric lymph node ↑ TNF, FoxP3, COX-2, and ß-catenin ↓ IFN-γ	[90]
C57BL/6 mouse	Lf^−/−^	AOM	DSS	Colonic dysplasia (31%) at 18 weeks	Acceleration of colitis ↑ IL-1ß, IL-6, CXCL1, MCP1, TNF-α, IFN-γ, NF-κB, and COX-2	[104]
C57BL/6 mouse	Nrf2^−/−^	AOM	DSS	Colonic tumors (93%) at 20 weeks and tumors consisted of adenomas (20%) and adenocarcinomas (80%)	Increased stromal nitrotyrosine-positive cells ↑ COX-2 (PGE_2_) and 5-LOX (LTB_4_)	[154]
C57BL/6 mouse	Syndecan-1^−/−^	AOM	DSS	Colonic adenomas (100%) with high-grade dysplasia at 9 weeks	Severe inflammation especially macrophage infiltration ↑ IL-6 and CCL2	[106]
C57BL/6 mouse	TLR2^−/−^	AOM	DSS	Large colorectal tumors, higher grade dysplasia (carcinoma *in situ*), and ACF at 9 weeks	Increased inflammatory cell infiltration ↑ IL-6, IL-17A, and phospho-STAT3	[155]
C57BL/6 mouse	No	AOM	TNBS	Aberrant crypts foci at 6 weeks	Severe colonic inflammation, mainly neutrophils and edema of submucosal and muscle layers	[87]
Wister rat	No	No	Ferric nitrilotriacetate	Renal adenoma or adenomatous hyperplasia (58%) and adenocarcinoma (54%) at 36 weeks	Acute nephrotoxicity in the proximal tubular epithelium and ROS generation by Fe^2+^ through Fenton reaction	[127]
F344 rat	No	No	Nickel subsulfide plus magnesium carbonate or iron	Renal mesenchymal tumors formed nickel alone (63%), nickel + magnesium (20%), and nickel + iron (60%) until 104 weeks	Nickel induced necrosis, inflammation (macrophages), fibrosis in the proximal tubular epithelium at the injection site	[128]
SKH-1 hairless mouse	Ptch1^+/−^	No	UVB	Basal cell carcinoma (100%) at 30 weeks	Recruitment of macrophages, neutrophils, mast cells, myeloid-derived suppressor cells (MDSCs), dendritic cells, NK cells, T cells, and B cells	[133]
Balb/c nude mouse	Xpa^−/−^	No	UVB	Skin tumor formation at 25 weeks	Neutrophil infiltration and ROS generation ↑ CXCL1, PGE_2_, TNF-α, and 8-OHdG	[134]

Abbreviations used are AOM, Azoxymethane; B[a]p, Benzo[a]pyrene; CCl_4_, carbon tetrachloride; DEN, N-diethylnitrosamine; DMBA, 7,12-Dimethylbenz[a]anthracene; DMH, 1,2-Dimethylhydrazine; DSS, Dextran sulfate sodium; HCC, hepatocellular carcinoma; iNOS; inducible nitric oxide synthase; LPS, lipopolysaccharide; MCA, 3-methylcholanthrene; NNK, nicotine-derived nitrosamine ketone [4-(methylnitrosamino)-1-(3-pyridyl)-1-butanone]; PDAC, pancreatic ductal adenocarcinoma; TNBS, 2,4,6-Trinitrobenzenesulfonic acid; and UVB, ultraviolet B.

**Table 2 cancers-13-00921-t002:** Surgery, hormone, or diet-induced inflammation-related carcinogenesis models.

Host/Strain	Genetic Manipulation	Carcinogen	Irritant or Manipulation	Incidence and Duration	Inflammatory Reaction	Reference
Sprague-Dawley rat	No	NNN	Esophageal duodenal anastomosis and iron	Esophageal adenocarcinoma (73%) at 30 weeks	Severe inflammation of the esophagoduodenal junction, as epithelial erosion and ulcer with infiltration of lymphocytes, eosinophils, and macrophages	[14]
Sprague-Dawley rat	No	No	Esophageal duodenal anastomosis	Esophageal adenocarcinoma (60%) at 35 weeks	Severe inflammation (lymphocytes, eosinophils, and macrophages) ↑ IL-1ß, IL-6, and IL-8	[156]
Wister rat	No	No	Gastroduodenostomy	Gastric carcinoma (71%) at 36 weeks	Pancreaticoduodenal secretions act as irritant and inducing inflammation	[36]
Wister rat	No	No	Gastrojejunostomy and NaHCO_3_	Gastric adenocarcinoma developed by supplementation with none (12%), CaHPO_4_2H_2_O (4%), NaCl (17%), NaHCO_3_ (54%), CaCO_3_ (61%) at 40 weeks	Inducing inflammatory reaction	[157]
Sprague-Dawley rat	No	DEN	Methionine-choline-deficient diet	Pre-neoplastic GST-P-positive foci from 2 weeks	Hepatic fibrosis with fat deposition and inflammation. Clusters of neutrophils/monocytes and ROS production ↑ NF-kB, IL-6, TGF-β, Col1a1, TIMP1, and TIMP2; ↓ IkB-alpha	[60]
FVB mouse	Myc TG	No	Methionine-choline-deficient diet	Liver tumors at 14 weeks	Selective loss of intrahepatic CD4^+^ but not CD8^+^ T cells. CD4^+^ T cells generate mitochondrially derived ROS	[59]
F344 rat	No	No	Choline-deficient, L-amino acid-defined diet	HCC (100%) until 96 weeks	Chronic inflammation, fibrosis, and cirrhosis ↑ TNF-α, IL-1-β, NF-kB, TGF-β, and COX-2	[61]
C57BL/6 mouse	No	DEN	High carbohydrate diet	HCC (100%) at 30 weeks	Hepatic inflammatory foci ↑ IL-1β, IL-18, Mcp1, Nalp3, JNK1/2, and ERK1/2	[40]
C57BL/6 mouse	No	No	High fat diet	HCC (42%) at 80 weeks	Lipid peroxidation and steatosis, NAFLD/NASH, fibrosis was developed, but no cirrhosis was formed	[62]
C57BL/6 mouse	No	DEN	High fat diet	HCC (100%) at 24 weeks	Hepatic inflammatory foci ↑ TNF-α, IL-1ß, IL-6, NF-kB, caspase-1, and STAT3; ↓ SIRT1 and AMPK	[63]
C57BL/6 mouse	No	DEN	Alcohol and Lieber-DeCarli alcohol diet	Hepatic hyperplasia at 13 weeks and alpha fetoprotein expression at 15 weeks	Infiltration of neutrophils and M2 macrophages and formation of liver fibrosis	[158]
Wister rat	Selectively bred for alcohol-preference	No	Lieber-DeCarli alcohol diet	Hepatic tumors (83%) at 72 weeks	Activation of Kupffer cells and increase oxidative stress	[54]
Sprague-Dawley rat	No	No	Ethanol	Oral cancer developed offspring male (30%) and female (39%) rats at 179 weeks	-	[9]
C57BL/6 mouse	No	4-NQO	Ethanol	Tongue squamous cell carcinoma (43%), severe dysplasia (5%), mild dysplasia (33%), and hyperplasia (19%) at 24 weeks	5-LOX and COX-2 positive inflammatory cells, especially mast cells were infiltrated	[11]
C57BL/6 mouse	No	DEN	Cholic acid	Hepatic adenoma, dysplastic nodules, and well-differentiated HCC at 40 weeks	Inflammatory cell infiltration ↑ TNF-α, IL-1-β, and NF-kB	[67]
Mouse	K-ras^G12D^	No	High-fat diet	PanIN-2 and PanIN-3 lesions in the pancreas formed at 10 weeks	High-fat diet increases inflammatory reaction including fibrosis, stromal collagen, macrophage infiltration, and K-ras activation ↑ COX-2	[78]
Pig	No	No	High-calorie diet	Early biomarkers of colon cancer risk, IL-6 and Ki-67 are increased at 13 weeks	Activation of IL-6 signaling (IL-1α, Akt3, PI3KR4, PIK3R5, MAPK10, and MAP2K2)	[94]
C57BL/6 mouse	No	No	Stress diet (High fat, low calcium, vitamin D, and methyl-donor nutrient)	Colorectal carcinoma (4%), tubulovillous adenoma (17%), tubular adenoma (8%), flat adenoma (13%) at 72 weeks	Eosinophils and basophils are infiltrated and formed nodules	[95]
Sprague-Dawley rat	No	DMBA	High-fat diet	Mammary tumors formed high corn oil-based diet (100%) and high olive oil-based diet (75%) at 35 weeks	Stromal reaction ↑ Il-1-α and TGF-β	[159]
Wister rat	No	No	Testosterone or prolactin	Testosterone- and prolactin-induced hyperplasia in ventral lobes and lateral/dorsal lobes, respectively	Prolactin induced leukocyte infiltration in prostate gland	[124]
NBL rat	No	No	Testosterone (T) plus estradiol-17β (E) or Diethylstilbestrol (D)	Prostate adenocarcinoma (83–100%), pituitary adenoma (100%) in T and E treated; Prostate adenocarcinoma (85–100%), pituitary adenoma (100%) in T and D-treated rats at 13 weeks	Inflammation in the lateral prostate lobes and pituitary organs	[160]
Sprague-Dawley rat	No	No	Testosterone (T) plus estradiol-17β (E) or Diethylstilbestrol (D)	Prostate adenocarcinoma (11–33%), pituitary adenoma (100%) in T and E treated; Prostate adenocarcinoma (17–66%), pituitary adenoma (100%) in T and D treated rats at 11 weeks	Inflammation in the lateral prostate lobes and pituitary organs	[160]
NBL rat	No	No	Testosterone plus estradiol-17β	Prostate adenocarcinoma (90%), pituitary tumor (100%), and mammary cancer (27%) at 54 weeks	Focal dysplastic and inflamed lesions were formed in the lateral prostate lobes by inflammation-mediated ROS	[123]

Abbreviations used are 4-NQO, 4-Nitroquinoline-1-oxide; NNN, N’-Nitrosonornicotine; and TG, transgenic.

**Table 3 cancers-13-00921-t003:** Genetically modified host-induced inflammation-related carcinogenesis models.

Host/Strain	Genetic Manipulation	Carcinogen	Irritant or Manipulation	Incidence and Duration	Inflammatory Reaction	Reference
C57BL/6 mouse	TLR2^−/−^	4-NQO	No	Atypical hyperplasia/carcinoma *in situ* (63%) and tongue carcinoma (25%) at 24 weeks	Macrophage infiltration ↑ IL-4, IL-6, IL-10, and IL-13	[12]
B6D2F_1_ mouse	IKKß Tg	No	No	Oral squamous cell carcinoma (10%) and carcinoma *in situ* (71%)	Oral carcinogenesis was caused by inflammatory granulocytes, macrophages, and B cells other than T cells, B cells, and NK cells	[8]
C57BL/6 mouse	human IL-1β TG	No	No	Columnar metaplasia (90%) before 60 weeks and high-grade esophageal dysplasia (20%) before 88 weeks	The inflammatory cell population contains CXCR4^+^ leukocytes (neutrophils and CD3^+^ T cells). No difference was observed in B cells, innate lymphoid cells, and NK-cells	[15]
129Sv-C57BL/6 mouse	Mig-6^d/d^K-ras^G12D^	No	No	Lung adenoma, atypical adenomatous, and hyperplasia within 16 weeks	Infiltration of WBC, lymphocytes, neutrophils, and macrophages ↑ CSF2, MIP1a, IL-13a1, TNFR2, COX2, and IL-18	[26]
Mouse	CCSP-rtTA/(tetO)_7_- Stat3C TG	No	No	Bronchoalveolar adenocarcinoma after 36 weeks	Lung inflammation including immune cell infiltration and up-regulation of proinflammatory cytokines and chemokines	[27]
B6D2F_1_ mouse	IKKß Tg	No	No	Forestomach squamous cell carcinoma (15%) and carcinoma *in situ* (15%)	Stomach carcinogenesis was caused by inflammatory granulocytes, macrophages, and B cells other than T cells, B cells, and NK cells	[8]
C57BL/6 mouse	Human IL-1β TG	No	No	Severe hyperplasia (>70%), high-grade dysplasia or well differentiated gastric adenocarcinoma (30%) over 48 weeks	Gastric chronic inflammation by myeloid-derived suppressor cells and independent of T cells and B cells ↑ β-catenin, c-myc, IL-6, TNF-α, and SDF-1α	[38]
Mouse	gp130^Y757F/Y757F^	No	No	Gastric adenoma at 16 weeks	Infiltrating lymphocytes in the lamina propria and the intraepithelial compartments	[37]
C57BL/6 mouse	TNAP-AID TG	No	No	HCC (27%), lung and stomach cancers (7%), and lymphoma (40%) at 90 weeks	AID expression induces inflammatory responses	[75]
Mouse	Β2-SP^+/-^	No	No	HCC (40–70%) at 60 weeks	Chronic activation of proinflammatory cytokines (IL-6 and TGF-β)	[70]
C57BL/6 mouse	Atg5^F/F^	No	No	Hepatocellular adenoma (100%) at 48 weeks	Increased apoptosis, inflammation, fibrosis, and infiltration of neutrophils and macrophages	[69]
C57BL/6 mouse	+Lepr^db^/ +Lepr^db^	DEN	No	Hepatic adenoma (70%) and HCC (10%) at 41 weeks	Chronic inflammation induced by lipids storage ↑ TNF-α, IL-1β, and IL-6	[73]
C57BL/6 mouse	mdr^−/−^	No	No	HCC (80%) at 64 weeks	M1 macrophages, TNF-α, NF-kB, and NO are increased; especially B cells are involved	[74]
129/OlaHsd mouse	mdr^−/−^	No	No	Liver nodules at 48 weeks	Nonsuppurative inflammatory cholangitis	[161]
C57BL/6 mouse	Fxr^−/−^	No	No	Liver tumors (38%) at 48 weeks. Histological type of HCC (40–70%) at 60 weeks. Pre-neoplastic foci (64%), adenoma (36%), HCC (6%), and mixed HCC & hepatocholangiocellular carcinoma (9%)	Hepatosteatosis-associated chronic inflammation and fibrosis ↑ IL- 1β and β-catenin	[71]
LEC rat	No	No	No	HCC (100%) at 72 weeks	Fulminant hepatitis induced by spontaneous hepatic copper & iron accumulation and its-mediated ROS generation	[47]
129SvEV mouse	APC^Min/+^IL-10^−/−^	No	No	Colon and cecal tumors at 20 weeks	Correlation between development colorectal cancer and extent of colon inflammation	[102]
FVB mouse	C3(1)SV40Tag TG	No	No	Mammary tumors (27%) at 21 weeks	Macrophage infiltration ↑ TNF-α, IL-6, IL-10, IL-12, MCP-1, and CD206	[111]
FVB mouse	C3(1)SV40Tag TG	No	No	Mammary intraepithelial neoplasia of low grade at 8 weeks, high grade at 12 weeks, and invasive mammary tumors at 16 weeks	-	[162]
CD1 mouse	neu^V^664^E^ TG	No	No	Mammary tumors (67%) until 21 weeks	-	[116]
Balb/c mouse	ERBB2 or HER-2/neu TG	No	No	Mammary in situ carcinomas and invasive cancer between weeks 17 and 22, palpable tumors at 33 weeks	Immune suppressive CD4^+^CD25^+^FOXP3^+^GITR^+^ T_reg_ cells and CD11b^+^Gr1^+^ cells are infiltrated	[117]
FVB mouse	PyV middle T antigen TG	No	No	Mammary adenocarcinoma (73%) until 25 weeks	-	[119]
C57BL/6 mouse	PyMT TG	No	No	-	Inflammatory monocyte-derived tumor-associated macrophage suppressed tumor-infiltrating CD8^+^ T cells	[120]
ACI/Seg rat	No	No	No	Prostatic cancer (>80%) at 144 weeks	Inflammation and aging-associated ROS	[163]
ACI/segHapBR rat	No	No	No	Intra-alveolar atypical hyperplasia (35–45%) at 96 weeks, prostatic atypical hyperplasia (95–100%), and invasive prostate carcinoma (35–40%) at 132 weeks	-	[164]
C57BL/6 mouse	Pten^fl/fl^	No	Expressions of IL-6 or oncostatin-M	Oncostatin-M expressing grafts developed poorly differentiated adenocarcinoma	Increased activation of JAK/STAT pathway	[125]
CD-1 mouse	Loss of XPA	No	No	Skin squamous cell carcinoma (100%) until 34 weeks	Accute inflammatory edema	[165]
C57BL/6 mouse	RacCA TG	No	No	Kaposi’s sarcomas in male mice at 32 weeks and in female mice at 72 weeks	↑ IL-6, IL-8, TNF-α, MCP-1, MIP1α, KC, and ROS	[142]

**Table 4 cancers-13-00921-t004:** Infection-induced inflammation-related carcinogenesis models.

Host/Strain	Genetic Manipulation	Carcinogen	Irritant or Manipulation	Incidence and Duration	Inflammatory Reaction	Reference
129Sv-C57BL/6 mouse	CCSP^Cre-Neo^/LSL–K-ras^G12D^	No	*Haemophilus influenzae*	Lung papillary adenoma and adenocarcinoma at 48 weeks	Neutrophil, macrophage, and CD8 T cells associated COPD-like airway inflammation	[21]
Mongolian gerbil	No	MNU	*Helicobacter pylori*	Gastric carcinoma (56%) and sarcoma (13%) at 75 weeks. Adenocarcinoma consisted of poorly differentiated (22%), signet ring cell (11%), and well-differentiated types (67%)	Active chronic gastritis, erosions, hyperplasia, and marked infiltration of inflammatory cells	[29]
C57BL/6 mouse	No	MNU	*Helicobacter pylori*	Gastric (antrum) adenoma (24%), adenocarcinoma (41%) comprised tubular adenocarcinoma (66%) and singlet-ring cell carcinoma (33%) at 50 weeks	Infiltrations of lymphocytes, plasma cells, neutrophils, and eosinophils were present in the lamina propria/submucosa and formation of lymphoid follicles ↑ TNF-α, TGF-ß, IL-6, IFN-γ, and LT-ß	[32]
C57BL/6 mouse	No	No	*Helicobacter felis*	Gastric adenocarcinoma (50%) at 12 months and 100% over 60 weeks	Approximately 4 months after infectious inflammation became prominent and seen as both submucosal and intramucosal infiltrates	[166]
C57BL/6 mouse	Rag2^−/−^	No	*Helicobacter pylori*	-	Gastroduodenitis formed extensive infiltration of the mucosa and submucosa with lymphocytes, macrophages, eosinophils, and neutrophils	[34]
Woodchuck	No	No	Woodchuck hepatitis virus	HCC (23%) around 236 weeks	Active hepatitis including neutrophils, eosinophils, lymphocytes, plasma cells, and histiocytes	[167]
C57BL/6 mouse	HCV core protein TG	No	No	Hepatic adenoma at 64 weeks and HCC (26–31%) until 76 weeks	Hepatic steatosis, lymphoid follicle formation, and eosinophilic cell infiltration ↑ TNF-α, IL-1β, MAPK, and ROS	[168]
C57BL/6 mouse	HBV HBsAg TG	No	No	HCC (91%) at 60 weeks	Activation of NF-κB pathway and inflammatory response ↑ ALT, AST, CD34, CD90, CD133, Sca1, Epcam, AFP, SOX9, and NF-κB	[43]
Syrian golden hamster	No	NDMA	*Opisthorchis viverrine*	Cholangiofibrosis and cholangiocarcinoma (100%) at 22 weeks	Eosinophils, neutrophils, or macrophages observed in the glandular lumen	[169]
Syrian golden hamster	No	NDMA	*Opisthorchis viverrine*	Cholangiocarcinoma (100%) at 24 weeks	Biliary epithelium is markedly inflamed ↑ IL-6	[52]
Syrian golden hamster	No	No	*Opisthorchis felineus*	Biliary intraepithelial neoplasia at 48 weeks	Granulomatous inflammation, monocytes, and eosinophil infiltration in the portal area in response to entrapped parasite eggs	[50]

Abbreviations used are MNU, N-methyl-N-nitrosourea and NDMA, N-nitrosodimethylamine.

**Table 5 cancers-13-00921-t005:** Foreign body-induced inflammation-related carcinogenesis models.

Host/Strain	Genetic Manipulation	Carcinogen	Irritant or Manipulation	Incidence and Duration	Inflammatory Reaction	Reference
C57BL/6 mouse	No	NDMA	Silica	Lung adenocarcinoma (>60%) at 48 weeks	Infiltration of T cells, B cells, neutrophils, and monocyte/macrophage cells	[170]
Fisher F344/NCr rat	No	No	Silica	Lung tumor (90%) composed of adenocarcinoma (84%), mixed carcinoma (8%), and squamous cell carcinoma (8%) at 68 weeks	Formation of silica granulomas, which consist of aggregates of activated macrophages and lymphoid cells ↑ ILs, TNF-α, TGF-ß, and ROS/NO	[18]
Balb/c mouse	No	NDMA	Silica	Lung adenoma (75%) at 26 weeks and adenocarcinoma (70%) at 48 weeks	Chronic inflammation ↑ TGF-ß, PD-1, LAG3, MCP-1, and FOXP3	[22]
F344 rat	No	No	Libby amphibole asbestos	Bronchiolar/alveolar adenoma or carcinoma (8%) at 31 weeks	Neutrophils and macrophages are predominantly infiltrated ↑ IL-6, IL-18, TNF-α, CXCL1, CXCL2, pAkt, pMEK1/2, and pSTAT3 ↓ IFN-γ	[17]
F344 rat	No	No	Amosite asbestos	Bronchiolar/alveolar adenoma or carcinoma (4%) at 31 weeks	Neutrophil infiltration ↑ IL-1ß, TNF-α, and CXCL1	[17]
Sprague-Dawley rat	human c-Ha-ras 128 Tg	DHPN	Titanium dioxide	Lung alveolar hyperplasia (100%) and adenoma (36%) at 16 weeks	Aggregates of TiO_2_ were localized exclusively in alveolar macrophages ↑ MIP1α and 8-OHdG	[19]
KSN nude mouse	No	No	Plastic plate	Colonic adenoma cells converted into moderately differentiated adenocarcinoma (65%) at 21 weeks	Plate-elicited chronic inflammation especially stromal reaction, but not acute inflammation is needed for tumorigenic conversion	[108]
SHR rat	No	No	Plastic plate	Non-tumorigenic cells converted into tumorigenic mammary tumor (100%) until 12 weeks	Neutrophils and macrophages are infiltrated into the periphery of the implanted plastic plate and proliferation of fibroblasts with lymphoid cells	[113]
C57BL/6 mouse	No	No	Plastic plate	Nontumorigenic cells converted into tumorigenic sarcoma (58%) until 6 weeks	Early phase of inflammation	[136]
C57BL/6 mouse	p53^+/-^	No	Plastic plate	Sarcoma formed in untreated (20%) and implanted with plastic plate (79%) at 70 weeks	Implanted plastic plate were covered by fibrous tissue capsules with accumulation of inflammatory cells in the surrounding stromal tissues ↑ ROS and RNS	[143]
C57BL/6 mouse	No	No	Gelatin sponge	Nontumorigenic cells converted into tumorigenic sarcoma (59%) until 6 weeks	Gelatin sponge-reactive inflammatory cells converted non-tumorigenic cells into tumorigenic ones	[137]

Abbreviation used was DHPN, Di(2-hydroxypropyl)nitrosamine.

## Data Availability

No new data were created or analyzed in this study. Data sharing is not applicable to this article.

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
