# Peer review of "Inflammation-Related Carcinogenesis: Lessons from Animal Models to Clinical Aspects"

_cancers, 2021, doi:10.3390/cancers13040921_

Round 1

Reviewer 1 Report

This review of the literature is of great interest, but there are numerous critical issues that I report below:
The chapter on materials and methods is completely absent; in fact, the inclusion and exclusion criteria are missing. A report of the reviewed and excluded studies is missing. The reader should immediately benefit from a flow-chart that schematises how this manuscript is structured. That is, how many apparatuses are studied and which models are reported.
Table 1, as it is structured, is unreadable; it would be advisable to divide the table into several tables divided by type of apparatus studied eg. lung cancer, etc.
The comment that is made for each apparatus and for each stimulation model should schematically report information that can subsequently be used by the reader, e.g. average time per model to obtain the result, stimulation concentrations, etc.
Furthermore, due to the absence of a structured review of the literature, the authors have lost some important models, for example: lung cancer and exposure to asbestos.

Author Response

"Responses to the Reviewers’ Comments"

All of the page and line numbers shown below correspond to blue highlighted text in the revised manuscript.

To Reviewer #1

The chapter on materials and methods is completely absent; in fact, the inclusion and exclusion criteria are missing. A report of the reviewed and excluded studies is missing. The reader should immediately benefit from a flow-chart that schematises how this manuscript is structured. That is, how many apparatuses are studied and which models are reported.

We have added Materials & Methods and explained how to select literature on page 2, line 85 to page 3, line 95.

Table 1, as it is structured, is unreadable; it would be advisable to divide the table into several tables divided by type of apparatus studied eg. lung cancer, etc.

Table 1 was divided into 5 categories (“chemical or irritant”, “surgery, hormone or diet”, “genetically modified host”, “infection”, and “foreign body”) according to the cause of inflammation. The explanation and 5 tables are inserted from page 20, line 1012 to page 30, line 1033.

The comment that is made for each apparatus and for each stimulation model should schematically report information that can subsequently be used by the reader, e.g. average time per model to obtain the result, stimulation concentrations, etc.

The average observation period, the type and histology of tumor that occurs, and the main inflammatory reactions are described. For details of experimental conditions such as administration concentration, the literature is provided so that the corresponding papers can be referred to. Shown in all tables on page 20, line 1017 to page 30, line 1033.

Furthermore, due to the absence of a structured review of the literature, the authors have lost some important models, for example: lung cancer and exposure to asbestos.

We have added lung carcinogenesis model by asbestos. We have inserted these explanations into page 4, lines 176-184; page 29; and added literature (#13) corresponding to the description on page 33, lines 1174-1176.

Reviewer 2 Report

This review discusses carcinogenic inflammatory factors caused by infection with pathogens or the uptake of foreign substances from the environment into the body. Inflammation-related carcinogenesis is reported to be responsible for 20% of cancer-related deaths. The review concludes that there is a common etiology of organ-specific animal models that mimic human inflammation-related carcinogenesis which involves prolonged exudation of inflammatory cells.

Specific points.

  • The review lists extensive evidence for the role of inflammation in a variety of cancers. However, this review is essentially descriptive and it provides little insight into the mechanisms through which inflammation affects, tumor growth, metastasis, immune responses and cancer treatments. Greater attention to the beneficial effects versus maladaptive effects of inflammation on these aspects would greatly improve the impact of this review.
  • There is surprisingly not a full description of inflammatory breast cancer.
  • The role of the tumor micro-environment and its interactions with tumor progression and outcomes could be much more developed in the review.
  • The role of high fat feeding, obesity, diabetes and inflamed adipose tissue in the progression of various cancers should receive much more attention.
  • In the abstract the authors propose that “extra corporeal remove therapy of inflammatory cells as a fundamental cause of inflammation-related carcinogenesis”. This reviewer does not understand what is implied by this statement and suggests that it is reworded.

Author Response

"Responses to the Reviewers’ Comments"

All of the page and line numbers shown below correspond to blue highlighted text in the revised manuscript.

To Reviewer #2

 The review lists extensive evidence for the role of inflammation in a variety of cancers. However, this review is essentially descriptive and it provides little insight into the mechanisms through which inflammation affects, tumor growth, metastasis, immune responses and cancer treatments. Greater attention to the beneficial effects versus maladaptive effects of inflammation on these aspects would greatly improve the impact of this review.

We agree that the paper on the effects of inflammation on tumor cells and tumor tissue environment will be valuable and interesting. We wrote some reviews about these. This manuscript does not focus on the effects of inflammation on tumor cell malignancy, but on carcinogenesis models in which inflammation is definitely involved. Added explanations to remind readers of the purpose of this manuscript on page 2, lines 78-82.

There is surprisingly not a full description of inflammatory breast cancer.

As far as we searched the literature, no experimental model of inflammatory breast cancer was hit. The search method we used and the method of selecting the articles adopted in this manuscript are described in Materials & Methods on page 2, line 85 to page 3, line 95.

The role of the tumor micro-environment and its interactions with tumor progression and outcomes could be much more developed in the review.  The role of high fat feeding, obesity, diabetes and inflamed adipose tissue in the progression of various cancers should receive much more attention.

Added a description of the focus of this manuscript on page 2, lines 78-82.

In the abstract the authors propose that “extra corporeal remove therapy of inflammatory cells as a fundamental cause of inflammation-related carcinogenesis”. This reviewer does not understand what is implied by this statement and suggests that it is reworded.

Rewrote the part pointed out by the Reviewer on page 1, lines 37-39.

Reviewer 3 Report

The authors reviewed animal models of inflammation-related carcinogenesis. They thoroughly investigated about inflammation-related carcinogenesis from ABC to application. The manuscript is written really well and looks almost more than enough for publication except one point. The instruction manual for authors suggests the main text of review papers should include at least “two figures or tables”. They have only one big table in the text. I suggest dividing the present one big table into “chemically induced inflammation” and “infection induced inflammation” etc. Or they should put one more figure or table.   

Author Response

"Responses to the Reviewers’ Comments"

All of the page and line numbers shown below correspond to blue highlighted text in the revised manuscript.

To Reviewer #3

The authors reviewed animal models of inflammation-related carcinogenesis. They thoroughly investigated about inflammation-related carcinogenesis from ABC to application. The manuscript is written really well and looks almost more than enough for publication except one point. The instruction manual for authors suggests the main text of review papers should include at least “two figures or tables”. They have only one big table in the text. I suggest dividing the present one big table into “chemically induced inflammation” and “infection induced inflammation” etc. Or they should put one more figure or table.  

We appreciate your supportive comments. According to the Reviewer’s suggestion, Table 1 was divided into 5 categories (“chemical or irritant”, “surgery, hormone or diet”, “genetically modified host”, “infection”, and “foreign body”) according to the cause of inflammation. The explanation and 5 tables are inserted from page 20, line 1012 to page 30, line 1033.

Round 2

Reviewer 2 Report

There was only a superficial attempt to address the short comings of this review.

Item 1. Page 2, lines  78-82. The authors could help the readers by giving citations to several reviews that have been published in the last two years on the role of inflammation and cancer progression.

Item 2. Inflammatory breast cancer still exist and could be discussed.  The fact that it was not picked up in the search perhaps reflects on the search criteria.

Item 3. This was not addressed.

Item 4 has been addressed.

Author Response

All of the page and line numbers shown below correspond to blue highlighted text in the re-revised manuscript.

To Reviewer #2

There was only a superficial attempt to address the short comings of this review.

Item 1. Page 2, lines  78-82. The authors could help the readers by giving citations to several reviews that have been published in the last two years on the role of inflammation and cancer progression.

We have added 5 review papers (#3-7) and changed the description on page 2, lines 78-82. We also changed the number of search articles on page 3, lines 92-96.

Item 2. Inflammatory breast cancer still exist and could be discussed.  The fact that it was not picked up in the search perhaps reflects on the search criteria.

We added inflammation-related mammary carcinogenesis models. We inserted explanations into page 17, lines 850-866; page 28; and added literature (#115-119).  

Item 3. This was not addressed.

For the general readers, review papers have been added on the metabolism-related inflammation that promote tumor progression (Reference #4-6). Explanations related to these have been added on page 2, lines 78-82.

Item 4 has been addressed.

Round 3

Reviewer 2 Report

We thank the authors for attending to outstanding criticisms.